# BANGS: GAME-THEORETIC NODE SELECTION FOR GRAPH SELF-TRAINING

**Fangxin Wang, Kay Liu, Sourav Medya, Philip S. Yu**
Department of Computer Science
University of Illinois Chicago
`fwang51, zliu234, medya, psyu@uic.edu`

## ABSTRACT

Graph self-training is a semi-supervised learning method that iteratively selects a set of unlabeled data to retrain the underlying graph neural network (GNN) model and improve its prediction performance. While selecting highly confident nodes has proven effective for self-training, this pseudo-labeling strategy ignores the combinatorial dependencies between nodes and suffers from a local view of the distribution. To overcome these issues, we propose BANGS, a novel framework that unifies the labeling strategy with conditional mutual information as the objective of node selection. Our approach—grounded in game theory—selects nodes in a combinatorial fashion and provides theoretical guarantees for robustness under noisy objective. More specifically, unlike traditional methods that rank and select nodes independently, BANGS considers nodes as a collective set in the self-training process. Our method demonstrates superior performance and robustness across various datasets, base models, and hyperparameter settings, outperforming existing techniques. The codebase is available on https://github.com/fangxin-wang/BANGS.

## 1 INTRODUCTION AND RELATED WORK

Self-training is a widely adopted strategy in semi-supervised graph learning (Lee et al., 2013) to transform a weak initial model into a stronger one (Frei et al., 2022). This approach enhances the base model's performance with reduced labeling effort and ensures the model's robustness against noisy or drifting data (Carmon et al., 2019; Wei et al., 2020). The iterative method is a common technique in self-training, where an initial teacher model—trained on labeled data—generates pseudo-labels for the unlabeled data. A student model is then trained to minimize empirical risk using both the labeled and pseudo-labeled data. Subsequently, the student model takes the role of the teacher, updating the pseudo-labels for the unlabeled data. This iterative process is repeated multiple times to progressively improve the final student model.

Existing research in graph self-training mainly focuses on solving three different aspects of the problem. *First, confirmation bias*, which refers to the phenomenon where the model overfits to incorrect pseudo-labels, is considered the main reason for the underperformance of naive self-training (Arazo et al., 2020). To tackle this issue in graph setting, current works generally rely on the confidence, i.e., probabilities of the most likely class, to select nodes for pseudo-labels. Common strategies include selecting the top $k$ nodes (Sun et al., 2020; Botao et al., 2023), or nodes with confidence surpassing a given threshold (Wang et al., 2021; Liu et al., 2022a). *Second, confidence calibration* refers to aligning the confidence of model predictions with the actual likelihood of those predictions being correct. Due to the neighborhood aggregation mechanism of Graph Neural Networks (GNNs), GNNs are often under-confident (Wang et al., 2021), which may lead to nodes with correctly predicted labels being excluded from the selection set. Therefore, GNN self-training is an important downstream task of GNN calibration, and the self-training performance is utilized as an evaluation metric of calibration (Liu et al., 2022b; Hsu et al., 2022; Li et al., 2023). *Third, information redundancy*, a concept from active learning (Li et al., 2024), has an important role in graph self-training. Nodes selected solely based on high confidence often carry similar information, which can lead to slow convergence and biased learning in the student model, even when the labels are correctly predicted. To address this, Liu et al. (2022a) select and pseudo-label high-confidence nodes

but reduce the weight of low-information nodes in the training loss to mitigate the redundancy in a post-hoc process. Similarly, as a pre-processing step, Li et al. (2023) employs mutual information maximization (Velickovic et al., 2019) to select more representative unlabeled nodes from their local neighborhoods.

All the works described above suffer from a few key limitations.
(1) *Non-unified methods:* Graph self-training is a sequential decision problem with multiple components, e.g., the selection and label criterion, the design of the teacher and the student model training framework. Due to the complexity of self-training problem, current works tend to treat these issues by proposing individual frameworks with different focuses.
(2) *Independent and iterative selection:* The self-training strategies require to use a set of points for pseudolabels. Most current graph self-training strategies only select nodes by ranking them independently based on some value such as confidence. The node selection process is a combinatorial problem, where the dependency between the candidate nodes is important in graphs.
(3) *Local view of distribution:* In each iteration, the student models learn from only labeled and pseudo-labeled data. Generalization over the entire distribution is hampered by the noisy or incorrect pseudo-label and possible distribution shifts.

**Our contributions.** Towards this end, we address these limitations by designing a framework that unifies several key insights for self-training. Our contribution can be summarized as follows.

- *Novel formulation.* To systematically address the graph self-training problem, we propose a novel objective for node selection based on the conditional mutual information. We demonstrate that this objective can be approximated using predictions on unlabeled samples, and leverage the feature influence from already labeled nodes to estimate the predictions of the student model for unlabeled samples.

- *Novel framework with theoretical support.* We introduce a game-theoretic framework, BANGS, which incorporates our designed objective as a utility function and select nodes in a combinatorial manner. We also show the limitations and non-optimality of independent selection, meanwhile providing theoretical support that BANGS can optimally select node sets even under a noisy utility function.

- *Experiments.* Experimental results validate the effectiveness of BANGS across various datasets and base models. By theoretically linking random walk and feature propagation, we enhance the scalability of our approach. Additionally, we demonstrate the effectiveness of BANGS under noisy labels and varying portion of training data.

## 2 PROBLEM DEFINITION

Consider a graph $\mathcal{G}$ containing a size-$N$ node set $\mathbb{V}$ where $v_i$ denotes the $i$-th node in $\mathcal{G}$. $\boldsymbol{X}$ and $\boldsymbol{A}$ denote the feature and adjacency matrix respectively in $\mathcal{G}$, where $\boldsymbol{x}_i$ represents the feature vector of the $i$-th node. $A_{i,j}$, the $i$-th row and $j$-th column element in $\boldsymbol{A}$, represents whether $i$-th node and $j$-th node is connected by an edge. We consider three different sets of nodes based on the labels: labeled nodes $\mathbb{V}^l$, pseudo-labeled nodes $\mathbb{V}^p$, and unlabeled nodes $\mathbb{V}^u$. We denote the ground truth and pseudo-labels of node $v_i$ as $y_i$ and $\hat{y}_i$ respectively. The node labels are from a set of $C$ classes, i.e., $y_i, \hat{y}_i \in \{1, 2, ..., C\}$. Table 4 provides an overview of the frequently used notations in this paper along with their corresponding explanations.

**Self-training.** In the iterative self-training procedure, the idea is to start with training a teacher model $f_0$ (at 0-th step) with $\mathcal{G} = \{\boldsymbol{A}, \boldsymbol{X}, \boldsymbol{y}^l\}$, where $\boldsymbol{y}^l$ denote the ground truth labels of $\mathbb{V}^l \subset \mathbb{V}$. Initially, at 0-th step, the set of nodes with pseudo labels is empty, i.e., $\mathbb{V}_0^p = \emptyset$. The teacher model $f_0$ generates pseudo-labels for all nodes in $\mathbb{V}_0^u$. At each iteration $r \in \{1, ..., R\}$, a subset of the unlabeled set $\mathbb{V}_{r-1}^u$ in the previous step, $\mathbb{V}_r^{p'}$ is selected based on a specific criteria. Thus, the pseudo-labeled set at $r$-the step becomes $\mathbb{V}_r^p = \mathbb{V}_{r-1}^p \cup \mathbb{V}_r^{p'}$, and unlabeled set $\mathbb{V}_r^u = \mathbb{V}_{r-1}^u \setminus \mathbb{V}_r^{p'}$. The pseudo-labels for these newly added samples are decided by the teacher model $f_{r-1}$. Subsequently, at the $r$-the step, the augmented $\mathbb{V}_r^p$ along with the originally labeled nodes $\mathbb{V}^l$ are used to train a new student model $f_r$, which uses additional pseudo-labels to improve its performance. In our setting, the student model $f_r$ in the $r$-th step is utilized as the teacher model in the $(r + 1)$-th step, allowing the framework to iteratively refine pseudo-labels based on the student model's improved understanding. We use step, round, and iteration interchangeably.

In previous self-training procedures, the goal for the student model $f_r$ at round $r$ is to minimize the loss function $\mathcal{L}_r(\mathbb{V}^l \cup \mathbb{V}^p_{r-1})$. This loss captures the classification error of $f_r$ in predicting only the currently available labels, including $\boldsymbol{y}^l$ for $\mathbb{V}^l$, and $\hat{\boldsymbol{y}}^p_{r-1}$ for $\mathbb{V}^p_{r-1}$. Note that $\boldsymbol{y}^l$ is the ground-truth labels, while $\hat{\boldsymbol{y}}^{p'}_s \subset \hat{\boldsymbol{y}}^p_{r-1}$ is predicted and determined by the corresponding teacher model $f_s$ for all $s \leq r-1$. Minimizing this loss function is considered as an approximation of the original goal which is to compute the prediction loss over the entire data distribution $V$, i.e., $\mathcal{L}_r(V)$, $\mathbb{V} \sim V$. However, the existing self-training frameworks use $\mathcal{L}_r(\mathbb{V}^l \cup \mathbb{V}^p_{r-1})$ as an intermediate approximate loss which often deviates from $\mathcal{L}_r(V)$ due to two major reasons: (i) the noisy pseudo-labels $\boldsymbol{y}^p_{r-1} \neq \hat{\boldsymbol{y}}^p_{r-1}$, and (ii) the non-random label selection strategy where the distribution of labeled nodes $\mathbb{V}^l \cup \mathbb{V}^p_{r-1}$ gradually shifts from the underlying distribution $V$. *To mitigate these, we design a framework where the student model learns from the pseudo-labels that are not only correct but also help in reducing the uncertainty of the model over the entire data distribution.*

**Entropy as a measure of uncertainty.** Shannon Entropy $H(\cdot)$ is commonly used to measure the uncertainty in a set of labeled nodes $\mathbb{S}$ and denoted as: $H(\mathbb{S}) = -\sum_{c=1}^{C} p_c \log p_c$, where $p_c$ is the probability of class $c$ on $\mathbb{S}$. Besides node set, we also use Entropy to measure the uncertainty of a single node $v_i$: $H(v_i) = -\sum_{c=1}^{C} p_c \log p_c$, where $p_c$ is the probability for the label of $v_i$ being class $c$ (Wang et al., 2024; Zou et al., 2025b). When $p_c$ of unlabeled nodes is estimated from the model predictions rather than being accessed from the ground-truth, $H(v_i)$ is refereed to as the *individual prediction entropy* of node $v_i$.

**Our problem.** We aim to quantify the following: *After selecting pseudo-label node set in the $r$-th iteration, how much information do we gain about the unlabeled data?* Formally, let $I(\mathbf{x}_1; \mathbf{x}_2) = H(\mathbf{x}_1) - H(\mathbf{x}_1|\mathbf{x}_2)$ denote the expected mutual information (information gain) between variables $\mathbf{x}_1$ and $\mathbf{x}_2$. In the $r$-th round, we choose only $k$ nodes to allocate pseudo-labels. Our aim is to select a node set $\mathbb{V}^{p'}_r$ of size $k$, such that the future pseudo-labels ($\hat{\boldsymbol{y}}^{p'}_r$) maximize information between $\boldsymbol{y}^u_{r-1}$, the ground-truth labels of unlabeled nodes and the prediction $\hat{\boldsymbol{y}}^u_{r-1}$ by the student model $f_r$. Mathematically, considering $y^u_{r-1}$ and $\hat{y}^u_{r-1}$, the ground-truth and prediction distribution of the unlabeled data respectively, our goal is as follows:

$$\max_{\mathbb{V}^{p'}_r \subset \mathbb{V}^u_{r-1}} \mathcal{O}(\hat{\boldsymbol{y}}^{p'}_r) = I(y^u_{r-1}; \hat{y}^u_{r-1}|\hat{\boldsymbol{y}}^{p'}_r, \hat{\boldsymbol{y}}^p_{r-1}, \mathcal{G})], \text{ s.t. } |\mathbb{V}^{p'}_r| = k. \tag{1}$$

**Lemma 2.1.** *Maximizing the mutual information between unlabeled data distribution and its prediction distribution is roughly equivalent to simultaneously maximizing entropy over the unlabeled dataset and minimizing the sum of individual prediction entropy over all unlabeled nodes.*

This lemma allows us to estimate unknown distribution with samples from data. Therefore, our new objective—which depends on entropy—becomes the following:

$$\max_{\mathbb{V}^{p'}_r \subset \mathbb{V}^u_{r-1}} \mathcal{O}(\hat{\boldsymbol{y}}^{p'}_r) \approx \frac{1}{|\mathbb{V}^u_{r-1}|} \sum_{v_i \in \mathbb{V}^u_{r-1}} H(\hat{y}_i|\hat{\boldsymbol{y}}^{p'}_r, \hat{\boldsymbol{y}}^p_{r-1}, \mathcal{G}) - H(\hat{y}^u_{r-1}|\hat{\boldsymbol{y}}^{p'}_r, \hat{\boldsymbol{y}}^p_{r-1}, \mathcal{G}),$$

$$\text{s.t. } |\mathbb{V}^{p'}_r| = k, \ \hat{y}^u_{r-1} = \frac{1}{|\mathbb{V}^u_{r-1}|} \sum_{v_i \in \mathbb{V}^u_{r-1}} \hat{y}_i. \tag{2}$$

The proof is in Appendix B.1. In other words, to let predicted labels have more information about ground-truth, the first term demands confident prediction on each sample, while the second term encourage diversity of prediction label distribution of unlabeled data. Solving this objective is non-trivial and has specific challenges as follows.

**C1.** The ground-truth distribution of labels $y^u_{r-1}$ is unknown for estimating the problem objective in Equation 1. We estimate it with the unlabeled samples by Equation 2.

**C2.** At the $r$-th round, we only have the predictions on the unlabeled data from the teacher model, $f_{r-1}$. In Equation 2, $\hat{y}^u_{r-1}$ represents the predictions made by the student model $f_r$, which may be inconsistent when compared to the teacher model's predictions, $f_{r-1}$, since the teacher model has not been trained on the newly added labels, $\hat{\boldsymbol{y}}^{p'}_r$. Thus, a key challenge is to estimate the student model's predictions based on the current teacher model efficiently and accurately (Section 3.1).

**C3.** How do we account for the dependency between $\hat{\boldsymbol{y}}^{p'}_r$, nodes that are being selected during each round? This dependency comes from the combinatorial nature of the objective and the graph—how much information the selected nodes *collectively* provide about the unlabeled ones (Section 3.2).

# 3 OUR METHOD: BANGS

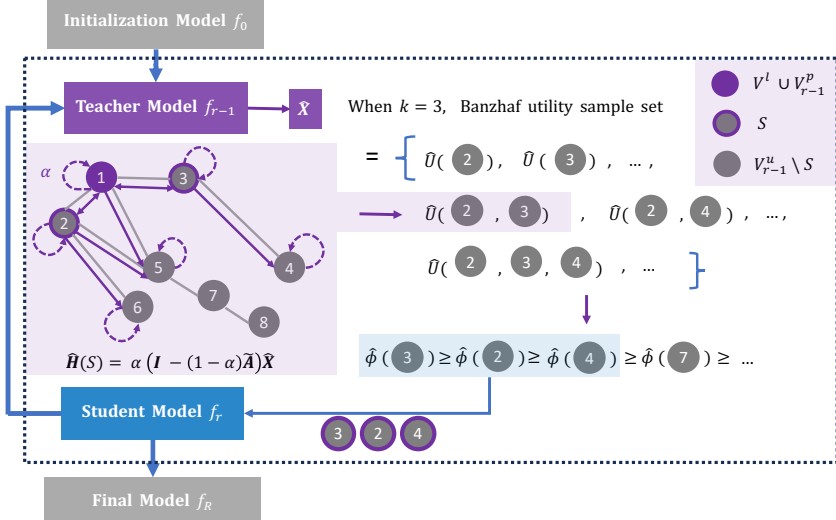

Figure 1: The workflow of BANGS. We utilize the teacher model's predictions to propagate features from both labeled nodes and candidate set $\mathbb{S}$, estimating the logits of unlabeled nodes and the Banzhaf value of set $\mathbb{S}$ (Equation 3). For simplicity, only the first step of feature propagation is shown. Using Banzhaf values we rank the individual contributions of unlabeled nodes and add the top $k$ into the pseudo-label set. The student model is subsequently retrained using this updated set.

In this section, we introduce our framework BANGS (**BAnzhaf value-based Node-selection for Graph Self training**), which consists of three major components unifying several concepts for self-training. *First,* we propose to measure the mutual information between labels and predictions through feature propagation from currently available labels and pseudo-labels to all the unlabeled nodes. *Second,* we apply Banzhaf values (Owen, 1975) from the co-operative game theory literature to compute the pseudo-label contribution for our self-training objective, taking into account the combinatorial dependencies of the nodes (Medya et al., 2020). *Third,* we provide scalability design and complexity analysis of BANGS. The pseudo-code and implementation details are in Appendix C and E.

## 3.1 FEATURE ESTIMATION WITH PROPAGATION

A major challenge in selecting node at round $r$ is having access to only the teacher model $f_{r-1}$ but not the student model $f_r$. An alternative is to estimate the true utility (information gain) of selecting the node set $\mathbb{V}_r^{p'}$ to train $f_r$ using the predictions from $f_{r-1}$. First, we introduce a feature propagation mechanism to estimate the unknown student model predictions accurately and efficiently.

In a $L$-layer GNN, labeled node $v_i$ can propagate its label information to $l$-hop neighbors, $l \leq L$ (Wang & Leskovec, 2020; Zou et al., 2025a). Therefore, pseudo-labeling node $v_i$ will influence the distribution and entropy of its neighbor nodes. In message-passing GNNs (Kipf & Welling, 2016; Velickovic et al., 2017), the influence of node $v_i$ on its neighbor $v_j$ depends on the graph structure. We measure the influence by computing how much the change in the input feature of node $v_i$, $\boldsymbol{x}_i^{(0)}$, affects the learned hidden feature of node $v_j$, i.e., $\boldsymbol{x}_j^{(l)}$ after $l$-step propagation (Xu et al., 2018b).

**Definition 3.1** (Feature influence distribution). The influence score of node $v_i$ on node $v_j$ after $l$-step propagation, $\hat{I}_f(i \rightarrow j, l)$, is defined as the sum of the absolute values of the expected Jacobian matrix $[|\mathbb{E}[\frac{\partial \boldsymbol{x}_j^{(l)}}{\partial \boldsymbol{x}_i^{(0)}}]|]$. The feature influence distribution of node $v_j$ is defined by normalizing over all nodes that affect node $v_j$ as follows: $I_f(i \rightarrow j, l) = \frac{\hat{I}_f(i \rightarrow j, l)}{\sum_{v_z \in \mathbb{V}} \hat{I}_f(z \rightarrow j, l)}$.

Under certain assumptions, the feature influence magnitude depends only on the graph structure and independent of the GNN model architecture, and it can be estimated via random walks.

*Assumption* 1. The $L$-layer GNNs have linear and graph structure dependent aggregation mechanism, and ReLU activation. All paths in the computation graph of the model are activated with the same probability of success.

**Theorem 3.1** (Feature influence computation via random walk). *With Assumption 1, $I_f(i \rightarrow j, L)$, the influence distribution of any node $v_i \in \mathbb{V}$ is nearly equivalent, in expectation to, the $L$-step random walk distribution on $\mathcal{G}$ starting at node $v_j$.*

The proof is in Appendix B.4. Theorem 3.1 implies that the more likely $v_j$ walks to $v_i$ after $L$ steps, the stronger raw features $\boldsymbol{x}_i^{(0)}$ will influence the final representation $\boldsymbol{x}_j^{(l)}$ in a $L$-layer GNN. During the forward propagation, node features are distributed across $L$ steps by the $L$-layer GNN, while the GNN is also trained via backpropagation. In this process, the gradient flows through the propagation mechanism, effectively accounting for an infinite number of neighborhood aggregation layers during the weight update. When $L \rightarrow \infty$, the random walk distribution is equivalent to personalized PageRank (PPR) (Gasteiger et al., 2018; 2022). Next, using PPR, we estimate the student model output by summing up the feature influence of all labeled $\mathbb{V}_l$, pseudo-labeled nodes $\mathbb{V}_{r-1}^p$, and the new selection set $\mathbb{V}_r^{p'}$ at round $r$.

**Definition 3.2** (Output feature estimation with propagation). With $\mathbb{V}_r^{p'}$ selected, the student model $f_r$ logit estimation $\hat{\boldsymbol{H}}(\mathbb{V}_r^{p'}) = \alpha(\boldsymbol{I} - (1-\alpha)\tilde{\boldsymbol{A}})^{-1}\hat{\boldsymbol{X}}$, where $\alpha \in (0, 1]$ is a teleportation probability, and $\tilde{\boldsymbol{A}} \in \mathbb{R}^{N \times N}$ is the normalized adjacency matrix with added self-loop. Further, $\hat{\boldsymbol{X}} \in \mathbb{R}^{N \times C}$ is a matrix that stores prediction logits of the teacher model $f_{r-1}$: for $v_j \in \mathbb{V}_l \cup \mathbb{V}_{r-1}^p \cup \mathbb{V}_r^{p'}$, the $j$-th row $\hat{\boldsymbol{X}}_{j,:}$ stores logits of $f_{r-1}$ for $v_j$; otherwise, $\hat{\boldsymbol{X}}_{j,:} = \boldsymbol{0}$.

We only propagate labeled and pseudo-labeled node logits to simulate back-propagation, as the loss function involves only the already labeled nodes in our framework. A theoretical case study on 1-layer GNN is provided in Appendix B.5. Based on this definition, we derive the final output feature estimation for the unlabeled nodes by student model $f_r$, which is subsequently processed through the softmax function (Equation 16) followed by the entropy function to produce the final entropy. Then, we can easily estimate the conditional mutual information between $\hat{y}_{r-1}^u$ and $y_{r-1}^u$ through Equation 2. This estimation captures the contribution of pseudo-labels to all unlabeled data, providing a global perspective and fast approximation without requiring model retraining. Relying on the relation between the PPR and the random walk, the output feature estimation can be implemented efficiently. A detailed complexity analysis is provided in Section 3.3.

## 3.2 MODELING INTERDEPENDENCIES THROUGH BANZHAF VALUES

To select a potentially good labeling set of nodes for self-training, it is crucial to compute the importance of the set based on the individual contribution of the nodes towards the final classification objective. However, the confidence of a node computed by the model might be dependent on the confidence of other nodes (Liu et al., 2022b). Intuitively, as a generalization, adding a pseudo-labeled node to the labeling set would affect the contribution of its neighborhood nodes. We show that the independent node selection procedure does not provide the optimal solution for Equation 1 both theoretically (Appendix B.2) and empirically (Section 4.3).

To capture the combinatorial dependency of the contributions of the nodes, we adopt a game theory-based measure, namely Banzhaf value (Wang & Jia, 2023; Chhablani et al., 2024). It assesses the contribution of each player (node) to the success of a coalition (node set), by measuring how pivotal their presence is for achieving a specific outcome, i.e., accuracy improvement in the next round. As we only select a set of size $k$, we define a variant of Banzhaf values where we consider the coalitions of size up to $k$:

**Definition 3.3** (k-Bounded Banzhaf value). The k-Bounded Banzhaf value for node $i \in \mathbb{V}_{r-1}^u$ with utility function $U(\mathbb{S})$ is defined as:

$$\phi(i; U, \mathbb{V}_{r-1}^u) := n_s^{-1} \sum_{\mathbb{S} \subseteq \mathbb{V}_{r-1}^u \setminus v_i, |\mathbb{S}|=m-1} [U(\mathbb{S} \cup v_i) - U(\mathbb{S})], \tag{3}$$

where $n_s = \sum_{m=1}^{k} \binom{|\mathbb{V}_{r-1}^u|-1}{m-1}$ denotes the number of all possible coalitions up to size $k$.

Here $\mathbb{S}$ is a subset of nodes of $\mathbb{V}_{r-1}^u$ excluding node $v_i$, $\mathbb{S} \cup v_i$ denotes the union set of $\mathbb{S}$ and node $v_i$. $U(\mathbb{S})$ denotes the utility function of $\mathbb{S}$, which maps the set $\mathbb{S}$ to an importance score of the subset $\mathbb{S}$. Breaking down into individual nodes, the value of $\phi(i; U, \mathbb{V}_{r-1}^u)$ measures the marginal contribution of node $v_i$ and can be used to ranks all the candidate nodes.

**Designing utility function.** Next, we design a relevant utility function to measure the total contribution of pseudo-labels in $\mathbb{S}$. A common choice for the utility function $U(\mathbb{S})$ is the prediction accuracy on a random hold-out subset of labeled nodes, which is an approximation of the contribution of subset $\mathbb{S}$ to the overall accuracy on $\mathbb{V}$ (Wang & Jia, 2023). However, this utility function is often neither effective nor efficient. *First*, the self-training task allows for only a few labels for validation; and *second*, accessing accuracy requires either retraining the model or approximating the influence function, which is time-consuming due to the exponential number of possible combinatorial sets.

To reflect the contribution of a subset to the overall performance in the *next* round of the self-training procedure, we follow the definition from the previous section. Specifically,

$$\hat{U}(\mathbb{S}) = \mathcal{O}(Softmax(\hat{\boldsymbol{H}}(\mathbb{S}))) - \mathcal{L}_{CE}(Softmax(\hat{\boldsymbol{H}}(\mathbb{V}^l)), \boldsymbol{y}^l) \tag{4}$$

where in the first term, $\mathcal{O}$ is illustrated in Equation 2 and prediction logits of unlabeled nodes are estimated through Def 3.2. The second term captures the cross-entropy loss (see $\mathcal{L}_{CE}$ in Table 4) between estimated probability and true label for all labeled data, preventing the confirmation bias. The ground-truth utility function $U(\mathbb{S})$ is defined in the same form, except using ground-truth student model logits $\tilde{\boldsymbol{X}}$ and not $\hat{\boldsymbol{H}}$. $\hat{U}(\mathbb{S})$ measures the total information propagated to unlabeled nodes by already labeled nodes $\mathbb{V}_l \cup \mathbb{V}_{r-1}^p$ and pseudo-labels of sampled set $\mathbb{S}$.

**Robust ranking under noisy utility function**. Ideally a method should preserve the optimal ranking order of nodes even with noisy utility function $\hat{U}(\cdot)$. Here, the noise arises due to the difference between our estimation $\hat{\boldsymbol{H}}(\mathbb{S})$ and the ground-truth student model logits $\tilde{\boldsymbol{X}}$. Let $\phi$ and $\hat{\phi}$ denote the Banzhaf values from the ground truth utility $U(\cdot)$ and the approximated (or noisy) utility $\hat{U}(\cdot)$, respectively. In the following, we prove that our computation of Banzhaf values preserve the ranking of nodes under noisy utility function $\hat{U}(\cdot)$.

According to Def. 3.3, the difference between the Banzhaf value of node $v_i$ and $v_j$ is given by $D_{i,j}(U) = k(\phi(i) - \phi(j))$. $U$ and $\hat{U}$ will produce reverse order of Banzhaf value for node $v_i$ and $v_j$ if and only if $D_{i,j}(U)D_{i,j}(\hat{U}) \leq 0$. We can view $U \in \mathbb{R}^{n_s}$ as a vector that maps all coalitions to corresponding utility values. Further, using the definition of $\phi$ in Def. 3.3, we can rewrite the following:

$$D_{i,j}(U) = \frac{k}{n_s} \sum_{m=1}^{k-1} \binom{|\mathbb{V}_{r-1}^u| - 2}{m - 1} \Delta_{i,j}^{(m)}(U), \tag{5}$$

where $\Delta_{i,j}^{(m)}(U) := \binom{|\mathbb{V}_{r-1}^u|-2}{m-1}^{-1} \sum_{\mathbb{S} \subseteq \mathbb{V}_{r-1}^u \setminus \{v_i, v_j\}, |\mathbb{S}|=m-1} [U(\mathbb{S} \cup v_i) - U(\mathbb{S} \cup v_j)]$ represents the average distinguishability between any unlabeled node $v_i$ and $v_j$ on size $m$ selection sets using the original utility function $U$.

In the next theorem, we show that when the value of utility between adding $v_i$ and $v_j$ is sufficiently large, and the estimation error in our utility function is moderate, the ranking of $v_i$ and $v_j$ remains robust even under noisy utility estimation $\hat{U}$. The proof is in Appendix B.6.

**Theorem 3.2.** *When the distinguishability of ground-truth utility between $v_i$ and $v_j$ on coalitions less than size $k$ is large enough, $\min_{m \leq k-1} \Delta_{i,j}^{(m)}(U) \geq \tau$, and the perturbation in utility functions is small such that $\left\| \hat{U} - U \right\|_2 \leq \tau \sqrt{\sum_{m=1}^{k-1} \binom{|\mathbb{V}_{r-1}^u|-2}{m-1}}$, then $D_{i,j}(U)D_{i,j}(\hat{U}) = (\hat{\phi}(i) - \hat{\phi}(j))(\phi(i) - \phi(j)) \geq 0$.*

**Ranking nodes for self-training selection**. Recall that our goal is to select a set of $k$ nodes that maximizes the objective in Equation 2, approximated by our utility function $\hat{U}$. Next we guarantee that our selection of top-k nodes even with $\hat{U}$ is within an error bound. Let us denote **Top**$(\phi, k)$ as the top $k$ node set ranked with $\phi$. Based on our objective, we only require

$D_{i,j}(U)D_{i,j}(\hat{U}) \geq 0$ for top-$k$ node $v_i$ and non-top-$k$ node $v_j$. According to Theorem 3.2, besides $\left\| \hat{U} - U \right\|_2 \leq \tau \sqrt{\sum_{m=1}^{k-1} \binom{|\mathbb{V}_{r-1}^u|-2}{m-1}}$, we only need $\min_{m \leq k} \Delta_{i,j}^{(m)}(U) \geq \tau$ for node pairs in $\{(v_i, v_j)|v_i \in \textbf{Top}(\phi, k), v_j \notin \textbf{Top}(\phi, k)\}$ to ensure $\textbf{Top}(\phi, k) = \textbf{Top}(\hat{\phi}, k)$. This is a less stricter condition, which only requires the utility function to distinguish between selected nodes and non-selected nodes, instead of any two nodes.

In conclusion, our Banzhaf value is tailored for the node self-training setting: 1) it adjusts the threshold for the size of coalitions based on the number of pseudo-labels to add, and 2) it relaxes the condition for ranking preservation without requiring comparison between all pairs of nodes.

### 3.3 COMPLEXITY ANALYSIS

We use Maximum Sample Reuse (MSR) Monte Carlo estimation (Wang & Jia, 2023) to calculate Equation 3 to speed up node selection. In expectation, $\phi(i; U, V) = n_s^{-1}(\mathbb{E}_S[U(\mathbb{S} \cup v_i)] - \mathbb{E}_S[U(\mathbb{S})])$, where $S \sim \mathbb{S} \subseteq \mathbb{V}_{r-1}^u \setminus v_i, |\mathbb{S}| = m - 1$. Draw $B$ samples of node sets independently from distribution, denoted as $\mathcal{S} = \{\mathbb{S}_1, ..., \mathbb{S}_b\}$. In terms of node $v_i$, the samples could be categorized into two classes: the samples including $v_i$, $\mathcal{S}_{\in i} = \{\mathbb{S} \in \mathcal{S} : v_i \in \mathbb{S}\}$; and the samples not containing $v_i$, $\mathcal{S}_{\notin i} = \{\mathbb{S} \in \mathcal{S} : v_i \notin \mathbb{S}\}$. Then $\phi(i; U, \mathbb{V})$ can be approximated by $\hat{\phi}_{MSR}(i; U, \mathbb{V}) = \frac{1}{|\mathcal{S}_{\in i}|} \sum_{S \in \mathcal{S}_{\in i}} U(\mathbb{S}) - \frac{1}{|\mathcal{S}_{\notin i}|} \sum_{S \in \mathcal{S}_{\notin i}} U(\mathbb{S})$.

Further, the utility function design involves PPR matrix. The computation of inverse $\hat{\boldsymbol{A}}^{-1}$ in Def. 3.2 is difficult in space for large graph. We instead use $H$-step power iteration, $\hat{\boldsymbol{X}}^{(h)} = (1 - \alpha)\hat{\boldsymbol{A}}\hat{\boldsymbol{X}}^{(h-1)} + \alpha\hat{\boldsymbol{X}}^{(0)}$ for $h \leq H$, which is initialized by $\hat{\boldsymbol{X}}^{(0)} = \hat{\boldsymbol{X}}$. The $L$-layer representation $\hat{\boldsymbol{H}}(\mathbb{V}_r^{p'}) := \hat{\boldsymbol{X}}^{(H)}$ is the final estimation.

*Total Time and Space Complexity*. We provide a detailed analysis in Appendix C. The final running time mainly depends on the selection of GNN architecture (e.g., type, the number of layers) in model retraining, along with Banzhaf sampling number and selected node set size. In practice, the model retraining step usually will take more time than the node selection step. Given $R$ self-training rounds, the total complexity is $O(R|\mathbb{V}|(H(|\mathbb{V}| + Bk)C + Ln_u n_e |\mathbb{V}|))$ where $H, B, L, n_u, n_e$ denote the PPR power iteration number, the sample number, and the number of layers, hidden units, and training epochs in training model. We also provide running time analysis in Appendix E.

The space bottleneck mainly lies in storing and multiplying $\hat{\boldsymbol{A}}$ with intermediate state $\hat{\boldsymbol{X}}^{(h-1)}$. Since the final logits are added influence from every available labels, it is suitable for batch computing. Further, by storing with a sparse matrix, the space complexity at most $O(|E^b||V^b|C)$ at the $r$-th iteration, where $V^b$ means the batch of labeled nodes, and $E^b$ denotes edges connected to $V^b$ in $\hat{\boldsymbol{A}}$.

## 4 EXPERIMENTS

In this section, we conduct experiments on 1) comparing our methods and baselines on the base GNN model across different datasets; 2) ablation studies; 3) hyperparameter analysis on the number of candidate and selected nodes; 4) robustness under noisy labels, different portions of training data, and different base GNN models. The experimental results demonstrate the effectiveness of our framework, particularly the integration of Banzhaf values and calibration.

### 4.1 SETTINGS

We test baseline methods and our method on five graph datasets: for Cora, Citeseer, and PubMed (Yang et al., 2016), we follow their official split; as for LastFM (Rozemberczki & Sarkar, 2020), Flickr (Zeng et al., 2019), we split them in a similar portion that training, validation, and test data take 5%, 15%, and 80%, respectively. The base model is set to Graph Convolutional Network (GCN) (Kipf & Welling, 2016) by default, while we also include results for other GNN models. The performance is evaluated on the best prediction accuracy of official test data; otherwise, we use all of the 80% unlabeled test data for evaluation. For each experiment, we select 10 different seeds and display their mean and standard deviation values. The best and second-best results are emphasized in bold and with underlines, respectively. More implementation details are in Appendix D.

Table 1: Node classification accuracy (%) with graph self-training strategies on different datasets. Mean, standard deviation, and single-side t-test on our method and second-best methods are demonstrated. Significance levels of 0.10 and 0.05 are indicated by "*" and "**".

| Dataset | Baselines | | | | | | BANGS |
|---------|-----------|------|-------|--------|--------|-----|-------|
|         | **Raw** | **M3S** | **CaGCN** | **DR-GST** | **Random** | **CPL** | |
| **Cora** | 80.82±0.14 | 81.40±0.29 | 83.06±0.11 | 83.04±0.38 | 83.16±0.10 | 83.72±0.52 | **84.23±0.62*** |
| **Citeseer** | 70.18±0.27 | 72.00±0.21 | 72.84±0.07 | 72.50±0.26 | 73.38±0.13 | 73.63±0.19 | **73.96±0.29**** |
| **PubMed** | 78.40±0.11 | 79.21±0.17 | 81.16±0.10 | 78.10±0.39 | 79.48±0.32 | 81.00±0.24 | **81.60±0.34**** |
| **LastFM** | 78.07±0.31 | 79.49±0.43 | 79.60±1.02 | 79.31±0.55 | 79.42±0.07 | 80.69±1.11 | **83.27±0.48**** |
| **Flickr** | 49.53±0.11 | 49.73±0.20 | 49.81±0.28 | 49.67±0.10 | 50.10±0.18 | 50.02±0.22 | **50.23±0.25** |

*Baselines.* We compare our methods with the following baselines.
1) **Raw** GNN is the base model without self-training; 2) **M3S** (Sun et al., 2020) uses deep-clustering to label nodes and selects top $k$ confident nodes; 3) **CaGCN** (Wang et al., 2021) calibrates confidence and selects nodes surpassing a pre-defined threshold; 4) **DR-GST** (Liu et al., 2022a) selects nodes surpassing a given threshold and reweights pseudo-labels in the loss of training student models with information gain; 5) **Random** selection shares the same hyperparamters with CPL Botao et al. (2023) and the nodes are selected randomly; 6) **CPL** (Botao et al., 2023) computes the multi-view confidence with dropout techniques and selects the top $k$ confident nodes. We employ cross entropy loss Cover et al. (1991) on the already pseudo-labeled and labeled data as the loss function in model re-training for all methods. For a fair comparison, we select the suggested hyperparameters for all baseline methods, especially in the node selection criterion. For instance, we use the suggested confidence threshold by CaGCN, e.g., 0.8 for Cora and 0.9 for Citeseer. We set the max iteration number as 40, and use validation data to early stop. For node selection, we sample 500 times for calculating Banzhaf values. The two varying hyperparameters are the number of candidate nodes $K$ and selected nodes $k$ in each iteration. The value of $k$ is set as 100 for small-scale graphs, i.e., Cora, Citeseer and PubMed, and 400 for other larger graphs; $K = k + 100$.
*Calibration.* Confidence calibration refers to aligning the confidence with the prediction accuracy, such that high confidence nodes have correct labels at a high probability. Before node selection, we apply confidence calibration to reduce noise in pseudo-labels and alleviate confirmation bias (Wang et al., 2021; Radhakrishnan et al., 2024). Further, calibration can reduce noise in utility function estimation, which is based on output logits propagation. Here, we use ETS (Zhang et al., 2020) for calibration, except CaGCN for PubMed. We also provide a case study on calibration in App. E.

## 4.2 Performance Evaluation

We select a 2-layer GCN as the base model and compared the node classification accuracy using various strategies. Notably, previous methods do not clearly define a stopping criterion, instead using the number of iterations as a hyperparameter. Therefore, we provide two fair comparisons. (1) Table 1 presents the best test accuracy achieved within the same 40 iterations for all methods. (2) In Table 6 of Appendix E.1, we also report the final test accuracy in the last round based on the number of suggested iterations for baselines. For our method, we present the test accuracy obtained by early stopping using validation data. Additionally, we performed a t-test to determine whether the mean accuracy of our methods is significantly greater than that of the second-best method. Significance levels of 0.10 and 0.05 are marked with "*" and "**", respectively. Our methods consistently outperform other confidence-based methods across most of the datasets. This supports that our framework, BANGS—especially combining Banzhaf value and information gain—is effective in practice.

## 4.3 Ablation Studies

In this set of experiments, we explore how different parts of node selection design help enhance performance in Table 2. Note that for a fair comparison, all experiments are under the same self-training framework and parameter. We compare with 1) **Random** selection, and 2) **Conf(Uncal)**, selecting the top $k$ most confident ones using uncalibrated confidence. To show that our performance gain does not depend only on confidence calibration, we compare with both 3) **Conf(CaGCN)** (Wang et al., 2021) and 4) **Conf(GATS)** (Hsu et al., 2022), which also select top confident $k$ nodes but calibrate confidence with CaGCN and GAT, respectively. Compared with CaGCN, GATS advances

Table 2: Ablation study of prediction accuracy (%) on different datasets.

| Dataset | Random | Conf(Uncal) | Conf(CaGCN) | Conf(GATS) | BANGS(Uncal) | BANGS(No Banzhaf) | BANGS |
|---|---|---|---|---|---|---|---|
| Cora | 83.16±0.10 | 83.70±0.56 | 83.72±0.49 | 83.76±0.18 | 83.73±0.43 | 83.03±0.66 | **84.23±0.62** |
| Citeseer | 73.38±0.13 | 73.74±0.19 | 74.02±0.52 | **74.18±0.38** | 73.64±0.33 | 72.75±0.50 | 73.96±0.29 |
| PubMed | 79.48±0.32 | 80.00±0.36 | 80.13±1.16 | 80.25±0.37 | 80.13±0.34 | 78.42±0.61 | **81.60±0.34** |
| LastFM | 79.42±0.07 | 80.69±1.11 | 83.23±0.25 | 80.31±0.78 | 80.77±1.10 | 82.62±0.44 | **83.27±0.48** |
| Flickr | 50.10±0.18 | 50.02±0.22 | 50.00±0.25 | 49.90±0.32 | 50.01 ± 0.26 | 49.98±0.17 | **50.23±0.25** |

graph calibration method through identifying and addressing multiple miscalibration-related factors. Though GATS is not designed for graph self-traning task, we adapt it to the same framework similar to Conf(CaGCN); We also explore whether each component is contributing to the final combination by removing each component respectively: 4) **BANGS(Uncal)**: our strategy without calibration; 5) **BANGS(No Banzhaf)**: directly selecting the top $k$ informative nodes from top $k + 100$ confident ones; and 6) **BANGS**, the proposed strategy.

Banzhaf value is an indispensable part of our methods: without it, the overlap in information propagated by previously selected nodes would be ignored. This is supported by our theoretical evidence (please see Appendix B.2 and B.6) and the significant decrease in performance in our experiments. On the other hand, calibration is effective in lifting the accuracy of both confidence-based methods and our methods, preventing noisy pseudo-labels from misguiding the training process. Except in Citeseer, our method benefits from calibration in getting correct and informative pseudo-labels.

### 4.4 HYPERPARAMETER ANALYSIS

Figure 2a show analysis of two hyperparameters collectively. In the first set of experiments, we set $k$, the number of nodes selected in iteration as 0, 5, 20, 50, 100, 200, 300, 400, 500, respectively, where 0 equals to raw GCN. The candidate nodes for $k$ smaller than 100 is set as $2k$, otherwise $k + 100$. The best number of added labels is around 50, with a balance between exploit and exploration – fewer nodes discourages student model from effectively and efficiently learns pseudo-label information, while more nodes tend to bring noisy and biased information to misguide model before it generalizes well. Though our method is more sensitive to the number of selected nodes in each iteration, the performance always outperform raw GCN. In the second set, we fixed $k$ as 100, and $K$ as 100, 125, 150, 175, 200, 250, 300, 400, 500. The more candidate nodes allows more possibly informative nodes to be included, while also have the risk of including noisy labels. This concern seems unnecessary in our experiments on Cora – on one hand, nodes generally have high confidence and therefore high accuracy after calibration; on the other hand, though including some more incorrect pseudo-labels, our selected correct nodes can effectively propagate maximized information benefiting both current and future iterations.

### 4.5 ROBUSTNESS

**Noisy Data**. We randomly select $\sigma$ portion of the training and validation nodes, and flip each label to another uniformly sampled different labels. The test accuracy of raw GCN, CPL (confidence only), and our methods on Cora are shown in Figure 2b. Though the initial performance is close, our methods significantly remain more robust on more noisy data. Interestingly, with a small portion (5%) of noise, our model would generalize better than on clean labels.

**Different Portions of Data**. In this set of experiments, we randomly select $\beta\%$ portion of training data and 15% validation data and show the test accuracy of raw GCN, CPL, and our methods. Different from official split that selects equal number of labels across all classes, uniform sampling leads to imbalanced labels and decreased performance. In this case, information redundancy is more of a problem – confidence based methods tend to select majority class to pseudo-label. As shown in Figure 2c, our methods consistently outperform other methods, especially with fewer training labels.

**Different Base Models**. Note that we have assumed the base model to be a $L$-layer GNN with a graph structure dependent linear aggregation mechanism and ReLU activation. Besides GCN, this assumption is often not strictly satisfied, but we demonstrate that our strategy can still preserve its validity, as shown in Table 3. Specifically, The aggregation mechanism of GraphSAGE (Hamilton et al., 2017) sums neighboring node features followed by normalization based on the degree of the nodes. GAT (Velickovic et al., 2017) places different weights on neighbors during aggregation ac-

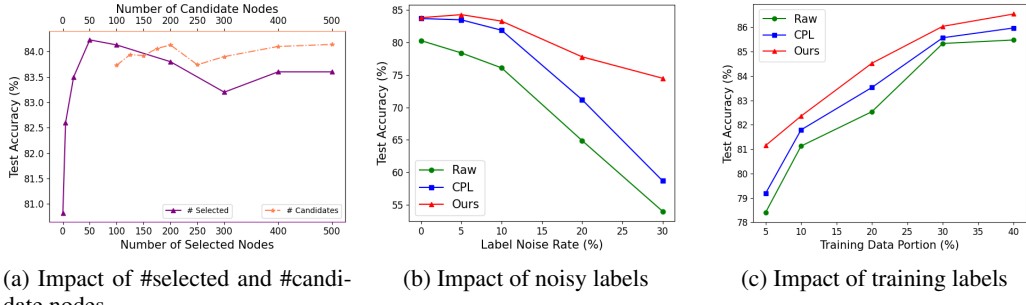

(a) Impact of #selected and #candidate nodes

(b) Impact of noisy labels

(c) Impact of training labels

Figure 2: Plots of hyperparameter and robustness analysis on Cora dataset. Our method retains the validity and superiority over baselines under different settings and hyperparameters.

Table 3: Test accuracy (%) on Cora dataset with different base models. Our method outperforms the baselines across different GNN models in most settings.

| Base Model | Baselines | | | | | | BANGS |
|---|---|---|---|---|---|---|---|
| | Raw | M3S | CaGCN | DR-GST | Random | CPL | |
| **GCN** | 80.82±0.14 | 81.40±0.29 | 83.06±0.11 | 83.04±0.38 | 83.16±0.10 | 83.72±0.52 | **84.23±0.62** |
| **GraphSAGE** | 81.50±0.03 | 82.86±0.72 | 83.04±0.36 | 83.50±0.21 | 83.01±0.38 | 83.43±0.15 | **83.94±0.32** |
| **GAT** | 81.13±0.32 | 82.80±1.88 | 81.70±0.28 | 83.33±1.14 | 83.10±0.22 | 83.65±0.35 | **83.86±0.50** |
| **GIN** | 77.47±0.57 | **81.35±0.21** | 80.75±0.44 | 78.56±0.55 | 78.72±0.37 | 80.01±0.51 | 80.77±0.68 |

cording to self-attention. Despite non-linearity of aggregation in GraphSAGE and GAT, our method preserves its best performance compared to baselines. With the base model as GIN (Xu et al., 2018a) with max aggregation, our method demonstrates the second-best performance.

## 5 CONCLUSIONS

In this paper, we have addressed the limitations in graph self-training by introducing a comprehensive framework that systematically tackles the node selection problem using a novel formulation with mutual information. We have proposed BANGS which is a game theory-based method with the utility function based on feature propagation. While BANGS exploits the combinatorial structure among the nodes, we have demonstrated the suboptimality of independent selection both theoretically and empirically. Additionally, we have shown that BANGS is able to preserve correct ranking even with noisy utility function. Extensive experiments validate the effectiveness and robustness of BANGS across different datasets, base models, and hyperparameter settings. These findings underscore the potential of our approach to advance graph-based learning models.

*Limitations and Future Work.* The setting of current self-training methods including ours has simplified assumptions. For example, the pseudo-labels selected in previous rounds are not considered to updated, and the teacher model in the next round are the same as the student model in the previous round. An interesting future direction would be designing methods under more general or different settings, e.g., on heterophilic (Chen et al., 2025) and large-scale graphs. Another future direction is to improve the efficiency of the utility function. One potential solution is through the use of influence functions, which, in current research, have been primarily explored on specific networks, e.g., GCNs (Chen et al., 2019; Kang et al., 2022). We believe extending this approach to more complex GNNs could lead us to a more efficient framework.

ACKNOWLEDGMENTS

This work is supported in part by NSF under grants III-2106758, and POSE-2346158.

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

# A NOTATION TABLE

Table 4: Frequently used notations and their explanations

| Notation | Explanation |
|---|---|
| $\mathcal{G}$ | Graph, $\mathcal{G} = \{\boldsymbol{A}, \boldsymbol{X}, \boldsymbol{y}^l\}$ |
| $\boldsymbol{A}$ | Adjacency matrix of graph $\mathcal{G}$ |
| $\boldsymbol{X}$ | Feature matrix of nodes |
| $r$ | Iteration round number |
| $\mathbb{V}^l$ | Set of labeled nodes |
| $\mathbb{V}^p_r$ | Set of all pseudo-labeled nodes |
| $\mathbb{V}^u_r$ | Set of unlabeled nodes in $r$-th round |
| $\mathbb{V}^{p'}_r$ | Set of newly added pseudo-labeled nodes in $r$-th round |
| $\boldsymbol{y}^l, \boldsymbol{y}^p_r, \boldsymbol{y}^u_r$ | Ground-truth Labels of labeled, pseudo-labeled and unlabeled nodes in $r$-th round |
| $f_r$ | Student model at $r$-th round, also equivalent to the teacher model at $(r+1)$-th round |
| $l$ | Influence propagation step number |
| $\hat{I}_f(i \to j, l)$ | The feature influence of node $v_i$ on node $v_j$ after $l$-steps |
| $H(v_i)$ | Entropy (uncertainty) of (in) a node $v_i$ |
| $\mathcal{O}(\mathbb{V}^{p'}_r)$ | The goal of self-training with $\mathbb{V}^{p'}_r$ selected in the $r$-th round |
| $\mathbb{S}$ | Sampled node set |
| $U(\cdot)$ | The noiseless utility function |
| $\hat{U}(\cdot)$ | The estimated utility function |
| $\phi(i; U, \mathbb{V})$ | Banzhaf value of node $v_i$ in candidate set $\mathbb{V}$ |
| $k$ | The number of selected nodes in each round |
| $K$ | The number of candidate nodes in each round |
| $\mathcal{L}_{CE}(\hat{\boldsymbol{y}}, \boldsymbol{y})$ | The cross-entropy loss between $\hat{\boldsymbol{y}}$ and $\boldsymbol{y}$, calculated by $-\frac{1}{|\boldsymbol{y}|}\sum_{i=1}^{|\boldsymbol{y}|}\sum_{c=1}^{C} y_{i,c}\log(\hat{y}_{i,c})$ |

# B PROOFS

## B.1 MUTUAL INFORMATION ESTIMATION

This section proves Lemma 2.1. Denote the distribution of unlabeled data as $y$ and its prediction as $u$. Following Bridle et al. (1991); Berthelot et al. (2019); Zhao et al. (2023), the objective of semi-supervised learning is formalized as

$$
\begin{aligned}
I(y; u) &= \iint p(y, u) \log \frac{p(y, u)}{p(y)p(u)} dy du \\
&= \int p(y) dy \int p(u|y) \log \frac{p(u|y)}{\int p(y)p(u|y)dy} du.
\end{aligned}
$$
(6)

Given we have $N$ samples. Writing with integrals, we obtain

$$
\begin{aligned}
I(y; u) =& \mathbb{E}_y\Big[\int p(u|y) \log \frac{p(u|y)}{\mathbb{E}_y[p(u|y)]} du\Big] \text{ (Taking expectation over } y) \\
=& \mathbb{E}_y\Big[\sum_i^N p(u_i|y) \log \frac{p(u_i|y)}{\mathbb{E}_y[p(u_i|y)]}\Big] \text{ (Taking expectation over } u) \\
=& \mathbb{E}_y\Big[\sum_i^N p(u_i|y) \log p(u_i|y)\Big] - \sum_i^N \mathbb{E}_y[p(u_i|y)] \log \mathbb{E}_y[p(u_i|y)] \text{ (Log rule)} \\
=& H(\mathbb{E}_y[f(u|y)]) - \mathbb{E}_y[H(f(u|y))].
\end{aligned}
$$
(7)

In the last line, the first term is entropy of the unlabeled dataset prediction distribution, and the second term equals to the sum of individual prediction entropy.

Next, we connect this lemma with our objective function design. In the $r-1$ round, $u$ is the prediction distribution of teacher model $f_{r-1}$, previous works that employ Equation 7 to optimize parameters of teacher prediction model $f_{r-1}$ in the current round. In our model, $u$ is the student model prediction distribution. While we fix the teacher model, and use Equation 7 to select nodes, such that the student model $f_r$ in the next round is improved. In a nutshell, we do not aim to optimize the current model but to let student model learn maximal mutual information about unlabeled data.

According to Equation 1, the goal can be approximated by an average over the unlabeled data set:

$$
\begin{aligned}
&I(y_{r-1}^u; \hat{y}_{r-1}^u | \hat{\boldsymbol{y}}_r^{p'}, \hat{\boldsymbol{y}}_{r-1}^p, \mathcal{G}) \\
&\approx \frac{1}{|\mathbb{V}_{r-1}^u|} \sum_{i=1}^{|\mathbb{V}_{r-1}^u|} \sum_{c=1}^{C} f_r(u) \log f_r(u) - \sum_{c=1}^{C} (\frac{1}{|\mathbb{V}_{r-1}^u|} \sum_{i=1}^{|\mathbb{V}_{r-1}^u|} f_r(u)) \log(\frac{1}{|\mathbb{V}_{r-1}^u|} \sum_{i=1}^{|\mathbb{V}_{r-1}^u|} f_r(u)) \\
&\approx \frac{1}{|\mathbb{V}_{r-1}^u|} \sum_{v_i \in \mathbb{V}_{r-1}^u} H(\hat{y}_i | \hat{\boldsymbol{y}}_r^{p'}, \hat{\boldsymbol{y}}_{r-1}^p, \mathcal{G}) - H(\hat{y}_{r-1}^u | \hat{\boldsymbol{y}}_r^{p'}, \hat{\boldsymbol{y}}_{r-1}^p, \mathcal{G}),
\end{aligned}
\tag{8}
$$

where $\hat{y}_{r-1}^u \approx \sum_{i=1}^{|\mathbb{V}_{r-1}^u|} f_r(u)$, is approximated by unlabeled data.

## B.2 WHY INDEPENDENT SELECTION IS NOT OPTIMAL?

In this section, we prove that independent selection is not optimal from two perspectives: the non-submodularity of the optimization problem, and the inequality of information gain.

By conditional information, our objective can be converted to

$$
\begin{aligned}
&I(y_{r-1}^u; \hat{y}_{r-1}^u | \hat{\boldsymbol{y}}_r^{p'}, \hat{\boldsymbol{y}}_{r-1}^p, \mathcal{G}) \\
&= I(y_{r-1}^u; \hat{y}_{r-1}^u, \hat{\boldsymbol{y}}_r^{p'} | \hat{\boldsymbol{y}}_{r-1}^p, \mathcal{G}) - I(y_{r-1}^u; \hat{\boldsymbol{y}}_r^{p'} | \hat{\boldsymbol{y}}_{r-1}^p, \mathcal{G}).
\end{aligned}
\tag{9}
$$

**Non-submodularity**. Interestingly, this objective function does not satisfy submodularity property, thus, a greedy algorithm does not guarantee approximation via submodularity. In graph learning, the logits are predicted by GNN fitted on node features, labels, and graph structure. Submodularity requires that for each $S \subset T$ and every node $v \notin T$, $U(S \cup v) - U(S) > U(T \cup v) - U(T)$. We can provide a simple counterexample to prove that the objective function is not submodular. Consider a graph with node $v_1, v_2, v_3, v_4$, and edge between $(v_1, v_2)$, $(v_2, v_3)$, and $(v_3, v_4)$. Node $v_2$ have two neighbors, Node $v_1$ and $v_3$. For simplicity, consider binary classification, node $v_1$, $v_2$, $v_4$ belong to class 0, and $v_3$ is class 1. Suppose an initial model can predict each node to the right class with 0.8. Let $S = \varnothing$, $T = \{v_1\}$, and $v = v_3$. When adding $v_3$ of class 1, the uncertainty over all other nodes will increase. This is because most GNNs have homophily assumption, and the other nodes are from a different class as $v_3$. Therefore, $U(S \cup v) - U(T)$ will be negative. When we have $T = \{v_1\}$, adding a $v_3$ from class 1 would still raise uncertainty over the unlabeled $v_2$ and $v_4$. But since GNNs aggregate information from neighbors, the influence from $v_3$ of different class is partially offset by $v_1$ of the correct class. Therefore, $U(T \cup v) - U(T) > U(S \cup v) - U(S)$.

**Inequality of Information**. Denote the pseudo-labels of $n$ nodes in $\mathbb{V}_r^{p'}$ as $\{\hat{y}_1^{p'}, \ldots \hat{y}_n^{p'}\}$. According to chain rule of mutual information, the first item

$$
\begin{aligned}
I(y_{r-1}^u; \hat{y}_{r-1}^u, \hat{\boldsymbol{y}}_r^{p'} | \hat{\boldsymbol{y}}_{r-1}^p, \mathcal{G}) =& I(y_{r-1}^u; \hat{y}_{r-1}^u, \hat{y}_1^{p'}, \ldots \hat{y}_n^{p'}, | \hat{\boldsymbol{y}}_{r-1}^p, \mathcal{G}) \\
=& I(\hat{y}_{r-1}^u, \hat{y}_1^{p'}; y_{r-1}^u | \hat{\boldsymbol{y}}_{r-1}^p, \mathcal{G}) + I(\hat{y}_{r-1}^u, \hat{y}_2^{p'}; y_{r-1}^u | \hat{y}_1^{p'}, \hat{\boldsymbol{y}}_{r-1}^p, \mathcal{G}) + \ldots \\
&+ I(\hat{y}_{r-1}^u, \hat{y}_n^{p'}; y_{r-1}^u | \hat{y}_1^{p'}, \ldots \hat{y}_{n-1}^{p'}, \hat{\boldsymbol{y}}_{r-1}^p, \mathcal{G})
\end{aligned}
\tag{10}
$$

While previous independent selection usually implictly optimize for a different goal:

$$
\max_{V_r^{p'} \subset \mathbb{V}_{r-1}^u} I(\hat{y}_1^{p'}; y_{r-1}^u | \hat{\boldsymbol{y}}_{r-1}^p, \mathcal{G}) + I(\hat{y}_2^{p'}; y_{r-1}^u | \hat{\boldsymbol{y}}_{r-1}^p, \mathcal{G}) + \cdots + I(\hat{y}_n^{p'}; y_{r-1}^u | \hat{\boldsymbol{y}}_{r-1}^p, \mathcal{G}),
\tag{11}
$$

or, considering the correlation with other labels,

$$
\begin{aligned}
\max_{V_r^{p'} \subset \mathbb{V}_{r-1}^u} \quad & I(\hat{y}_{r-1}^u, \hat{y}_1^{p'}; y_{r-1}^u | \hat{\boldsymbol{y}}_{r-1}^p, \mathcal{G}) + I(\hat{y}_{r-1}^u, \hat{y}_2^{p'}; y_{r-1}^u | \hat{\boldsymbol{y}}_{r-1}^p, \mathcal{G}) + \dots \\
& + I(\hat{y}_{r-1}^u, \hat{y}_n^{p'}; y_{r-1}^u | \hat{\boldsymbol{y}}_{r-1}^p, \mathcal{G}),
\end{aligned}
\tag{12}
$$

Specifically, the matched items in Equation 10 and 11 or 12 usually do **not** equal to each other. Take the $j$-th item and one node $v_i \in \mathbb{V}_{r-1}^u$ as an example. Their difference, namely **interation information** in information theory, is formulated as

$$
\Delta_j^1 = I(\hat{y}_{r-1}^u, \hat{y}_j^{p'}; y_{r-1}^u | \hat{y}_1^{p'}, \dots \hat{y}_{j-1}^{p'}, \hat{\boldsymbol{y}}_{r-1}^p, \mathcal{G}) - I(\hat{y}_{r-1}^u, \hat{y}_j^{p'}; y_{r-1}^u | \hat{\boldsymbol{y}}_{r-1}^p, \mathcal{G})
\tag{13}
$$

Nonetheless, by the non-monotonicity of conditional mutual information, conditioning can either increase, preserve, or decrease the mutual information between two variables (Vu, 2024). This is the case even when random variables are pairwise independent.

Similarly, we can derive the interaction information for the second item in Equation 9 that:

$$
\Delta_j^2 = -I(\hat{y}_j^{p'}; y_{r-1}^u | \hat{y}_1^{p'}, \dots \hat{y}_{j-1}^{p'}, \hat{\boldsymbol{y}}_{r-1}^p, \mathcal{G}) + I(\hat{y}_j^{p'}; y_{r-1}^u | \hat{\boldsymbol{y}}_{r-1}^p, \mathcal{G})
\tag{14}
$$

Only when $\sum_{j=1}^n (\Delta_j^1 + \Delta_j^2) = 0$, independent selection provides the optimal estimation of goal. However, this is not guaranteed in practice without further assumption. Therefore, a combinatorial selection method would be more effective than the independent selection procedure.

## B.3 PRELIMINARY: GRAPH NEURAL NETWORKS

To start proof, we first formally formulate Graph Neural Networks (GNN). For node $v_i$, GNNs utilize its feature $\boldsymbol{x}_i \in \mathbb{R}^D$ of $D$ size as the initial embedding $\boldsymbol{x}_i^{(0)}$ and adjacency matrix between $v_i$ and its neighborhood nodes $\mathcal{N}_i$. Similarly, GNNs use $A_{i,j}$ to create inital edge embedding $\boldsymbol{e}_i^{(0)}$. Denote the embedding in layer $l$ (i.e., after $l$-th propagation) as $\boldsymbol{x}_i^{(l)}$, and its value at the $d$'th position as $\boldsymbol{x}_{id}^{(l)}$. Commonly, GNNs can be described through:

$$
\begin{aligned}
\boldsymbol{x}_i^{(l)} =& f_{node}(\boldsymbol{x}_i^{(l-1)}, AGG_{v_j \in \mathcal{N}_i}[f_{msg}(\boldsymbol{x}_i^{(l-1)}, \boldsymbol{x}_j^{(l-1)}, A_{i,j})]), \\
\boldsymbol{e}_i^{(l)} =& f_{edge}(\boldsymbol{x}_i^{(l)}, \boldsymbol{x}_j^{(l)}, \boldsymbol{e}_i^{(l-1)}).
\end{aligned}
\tag{15}
$$

Here, $AGG[\cdot]$ is a function to aggregate the node and its neighborhood embeddings, and $f_{node}$, $f_{edge}$ and $f_{msg}$ are functions, which can be linear layers or models with skip connections.

Deonote the output logit value at dimension $d$ in the last $L$ layer as $x_{i,d}^{(l)}$. The final predicted probability of class $d$ is often calculated through softmax function:

$$
P_{i,d} = \text{Softmax}(x_{i,d}^{(L)}) = \frac{e^{x_{i,d}^{(L)}}}{\sum_{c=1}^C e^{x_{i,d}^{(L)}}}.
\tag{16}
$$

In a semi-supervised setting, we have access to ground-truth labels $\boldsymbol{y}^l$ of a few nodes $\mathbb{V}^l$. In back propagation, the labels influence the weight and bias of $f_{node}$ and $f_{edge}$ through gradient descent.

## B.4 CONNECTION BETWEEN FEATURE INFLUENCE AND RANDOM WALK

*Proof.* In a general neural network with ReLU activation, the $d_1$-th output logit can be written as $x_{d_1}^{(L)} = \frac{1}{\lambda^{(L-1)/2}} \sum_{q=1}^{\phi} z_{d_1,q} x_{d_1,q} \prod_{l=1}^L W_{d_1,q}^{(l)}$ (Choromanska et al., 2015). $\lambda$ is a constant related

to the size of neural network, and $L$ is the hidden layer number. Here, $\phi$ denotes the total number of paths, $z_{d_1,q} \in 0, 1$ denotes whether computational path $p$ is activated, $x_{d_1,q}$ represents the input feature used in the $q$-th path of logit $d_1$, and $W_{d_1,q}^{(l)}$ is the used entry of weight matrix at layer $l$. By assuming equal activation probability of $\rho$, we get:

$$\boldsymbol{x}_{d_1}^{(L)} = \frac{\rho}{\lambda^{(L-1)/2}} \sum_{q=1}^{\phi} \boldsymbol{x}_{d_1,q} \prod_{l=1}^{L} W_{d_1,q}^{(l)}. \tag{17}$$

Under Assumption 1, the $l$ layer feature updating equation in Equation 15 can be rewritten as

$$\boldsymbol{x}_i^{(l)} = ReLU(\boldsymbol{W}^{(l-1)} \cdot AGG_{v_j \in \mathcal{N}_i}[f_{msg}(\boldsymbol{x}_i^{(l-1)}, \boldsymbol{x}_j^{(l-1)}, A_{i,j})]), \tag{18}$$

where $\boldsymbol{W}^{(l)}$ is a trainable weight matrix in the $l$-th layer of GNN.

According to Gasteiger et al. (2022), from the view of path, the feature of starting node $i$ is influenced by ending node $j$ by all $\phi$ activated paths $q$ through the acyclic *computational* graph structure, derived from the GNN structure. The activation status of path $q$ depends on ReLU function. Therefore, $\boldsymbol{x}_{i,d}^{(l)}$ the $d$-th logit of $\boldsymbol{x}_i^{(l)}$, could also be expressed as

$$\boldsymbol{x}_{j,d}^{(l)} = \frac{1}{\lambda^{(L-1)/2}} \sum_{v_i \in \mathbb{V}} \sum_{p=1}^{\psi} \sum_{q=1}^{\phi} z_{i,d,p,q} x_{i,d,p,q} \prod_{l=1}^{L} a_{i,p}^{(l)} W_{d,q}^{(l)}, \tag{19}$$

Here, we introduce *data-based* graph path $q$, which is decided by $\mathcal{G}$. $z_{i,d,p,q} \in \{0, 1\}$ is a indicator of whether the graph path $q$ is active with the ReLU in the *computational* graph. Note that $p$ is counted to $\psi$ separately for each combination of the output(ending) node $v_i$, logit position $d$ and the *computational* graph path $q$. Similarly, define $x_{i,d,p,q}$ is the input feature of $v_i$ used in the $q$-th computational path, for graph path $p$ at output logit $d$. $a_{i,p}^{(l)}$ denotes the normalized graph-dependent aggregation weights of edges, parameter of $AGG[\cdot]$. $W_{d,q}^{(l)}$ is the entry for layer $l$ and logit $d$ in the weight matrix $\boldsymbol{W}$, decided by the GNN structure.

According to Definition 3.1, the $L$-step influence score of node $v_i$ on $v_j$ is computed by

$$\hat{I}_f(i \rightarrow j, L) = \sum_{d_1} \sum_{d_2} |\mathbb{E}[\frac{\partial x_{j,d_1}^{(L)}}{\partial x_{i,d_2}^{(0)}}]|. \tag{20}$$

Under a $L$-layer GNN, we calculate the derivative of Equation 19:

$$\frac{\partial x_{j,d_1}^{(L)}}{\partial x_{i,d_2}^{(0)}} = \frac{1}{\lambda^{(L-1)/2}} \sum_{p=1}^{\psi} \sum_{q=1}^{\phi'} z_{i,d_1,p,q} \prod_{l=1}^{L} a_{i,p}^{(l)} W_{d_1,q}^{(l)}, \tag{21}$$

where the $\boldsymbol{x}_{j,d_1}^{(L)}$ represents the output feature of $v_j$ at $d_1$ position, $\boldsymbol{x}_{i,d_2}^{(0)}$ represents the input feature of $v_i$ at $d_2$ position. $\phi'$ denotes the number of *computational* paths related to input dimension $d_2$ and output dimension $d_1$. In Assumption 1, all paths in the computation graph of the model are activated with the same probability of success $\rho$, i.e., $\mathbb{E}[z_{i,d_1,p,q}] = \rho$, then the expectation of Equation 21 can be written as follows. The second line in this equation holds, as $a_{i,p}^{(l)}$ only depends on the path in graph $\mathcal{G}$, independent of $W_{d,q}^{(l)}$.

$$\mathbb{E}[\frac{\partial x_{j,d_1}^{(L)}}{\partial x_{i,d_2}^{(0)}}] = \frac{1}{\lambda^{(L-1)/2}} \sum_{p=1}^{\psi} \sum_{q=1}^{\phi'} \rho \prod_{l=1}^{L} a_{i,p}^{(l)} W_{d_1,q}^{(l)}$$

$$= \frac{\rho}{\lambda^{(L-1)/2}} (\sum_{p=1}^{\psi} \prod_{l=1}^{L} a_{i,p}^{(l)})(\sum_{q=1}^{\phi'} \prod_{l=1}^{L} W_{d_1,q}^{(l)}). \tag{22}$$

Here, item $\sum_{p=1}^{\psi} \prod_{l=1}^{L} a_{i,p}^{(l)}$ sums over probabilities of all possible paths of length $L$ from node $v_j$ to $v_i$, which is also the probability that a random walk starting at $v_i$ and ending at $v_j$ after taking $L$ steps. Denote this probability as $P_L^{j \to i}$. Since this random walk item is independent of feature dimension $d_1$ and $d_2$, summing up Equation 22, we get another expression of Equation 20:

$$\hat{I}_f(i \to j, L) = \frac{\rho}{\lambda^{(L-1)/2}} P_L^{j \to i} (\sum_{d_1} \sum_{d_2} |\sum_{q=1}^{\phi'} \prod_{l=1}^{L} W_{d_1,q}^{(l)}|). \tag{23}$$

Note that $\boldsymbol{W}^{(l)}$, as trainable weight matrix in layer $l$, shares the same value for all nodes. Except $P_L^{j \to i}$, all other items are independent on nodes. Therefore, the expectation of $\hat{I}_f(i \to j, L)$, the influence distribution of any node $v_i \in \mathbb{V}$ is in proportion to the expectation of $L$-step random walk distribution on $G$ starting at node $v_j$. Since $I_f(i \to j, L) = \frac{\hat{I}_f(i \to j, L)}{\sum_{v_z \in \mathbb{V}} \hat{I}_f(z \to j, L)}$ is normalized over all nodes, the influence distribution is equivalent in expectation to random walk distribution. Formally, Equation 23 can be simplified to

$$I_f(i \to j, L) \sim P_L^{j \to i} \tag{24}$$

Theorem proved.

$\square$

### B.5 CASE STUDY: 1-LAYER GNN

#### B.5.1 BOUNDING ESTIMATION ERROR OF ENTROPIES BY LOGITS

**Lemma B.1.** *Given two distributions $P_1 = \{p_{1,d}\}$ and $P_2 = \{p_{2,d}\}$, the probabilities are obtained from logits using the softmax function in Equation 16, and assume the difference in logits $x_{1,d} = x_{2,d} + c_j$ for every dimension $d$. Then the final probability distribution $P_1$ is a reweighted version of $P_2$:*

$$p_{1,j} = \frac{e^{c_j}}{\sum_{k \in C} e^{c_k}} \cdot p_{2,j}.$$

*Proof.* To make $P_1$ and $P_2$ identical, we require $p_{1,j} = p_{2,j}$ for all $j \in C$. This implies:

$$\frac{e^{x_{1,j}}}{e^{x_{2,j}}} = \frac{\sum_{k \in C} e^{x_{1,k}}}{\sum_{k \in C} e^{x_{2,k}}}$$

Taking $\log$ on both sides:

$$x_{1,j} - x_{2,j} = \text{some constant } c$$

This shows that $x_{1,j}$ and $x_{2,j}$ must differ by the same constant across all classes $j$, i.e., for all $j \in C$, where $c$ is a constant independent of $j$.

Now, if the difference between the logits $x_{1,j}$ and $x_{2,j}$ is not constant but depends on the class $j$ as $c_j$, then we have:

$$x_{1,j} = x_{2,j} + c_j$$

In this case, the probabilities $p_{1,j}$ can be expressed as:

$$p_{1,j} = \frac{e^{x_{1,j}}}{\sum_{k \in C} e^{x_{1,k}}} = \frac{e^{x_{2,j}+c_j}}{\sum_{k \in C} e^{x_{2,k}+c_k}} = \frac{e^{c_j}}{\sum_{k \in C} e^{c_k}} \cdot p_{2,j}$$

Thus, the final probability distribution $P_1$ is a reweighted version of $P_2$, where each probability $p_{2,j}$ is scaled by a factor $e^{c_j}$ and then normalized across all classes $C$. $\qquad\square$

**Lemma B.2.** *When $c_j$ in Lemma B.1 is small enough, and for any class $i, j$, $|c_i - c_j| \le \delta$, then the absolute entropy difference between $P_1$ and $P_2$:*

$$|H(P_1) - H(P_2)| \le \frac{\delta}{2}.$$

*Proof.* According to definition of Shannon entropy,

$$H(P_1) - H(P_2) = -\sum_{j \in C} p_{1,j} \log p_{1,j} + \sum_{j \in C} p_{2,j} \log p_{2,j}.$$

By substituting with relationship between $p_{1,j}$ and $p_{2,j}$ in Lemma B.1, we get

$$H(P_1) - H(P_2) = -\sum_{j \in C} \left( \frac{e^{c_j}}{\sum_{k \in C} e^{c_k}} \cdot p_{2,j} \right) \left( c_j - \log \sum_{k \in C} e^{c_k} \right) = -\sum_{j \in C} p_{1,j} \cdot \left( c_j - \log \sum_{k \in C} e^{c_k} \right)$$

For small $c_j$, an approximation could be:

$$H(P_1) - H(P_2) \approx -\sum_{j \in C} p_{1,j} \cdot (c_j - \bar{c})$$

where $\bar{c}$ is the average of $c_j$. This shows that the entropy difference is a weighted sum of deviations of $c_j$ from its mean, weighted by the probabilities $p_{1,j}$.

Further, if for any class $i, j$, $|c_i - c_j| \le \delta$, then

$$|c_j - \bar{c}| \le \frac{\delta}{2}.$$

Therefore, the entropy estimation error

$$|H(P_1) - H(P_2)| \le \sum_{j \in C} p_{1,j} \frac{\delta}{2} = \frac{\delta}{2} \sum_{j \in C} p_{1,j} = \frac{\delta}{2}.$$

$\qquad\square$

### B.5.2 PREDICT THEN PROPAGATE

Comparing with Equation 22, we are interested in the logit value $\mathbb{E}[x_{j,d_1}^{(L)}]$, instead of derivative $\mathbb{E}[\frac{\partial x_{j,d_1}^{(L)}}{\partial x_{i,d_2}^{(0)}}]$. Therefore, the computational paths are not only those connected to input feature dimension at $d_2$ position and output at $d_1$, but all activated computational paths that output at $d_1$. Thus, from Equation 21, we get:

$$\mathbb{E}[x_{j,d_1}^{(L)}] = \frac{\rho}{\lambda^{(L-1)/2}} \sum_{v_i \in \mathbb{V}} \sum_{p=1}^{\psi} \sum_{q=1}^{\phi} x_{i,d_1,p,q} \prod_{l=1}^{L} a_{i,p}^{(l)} W_{d_1,q}^{(l)}. \tag{25}$$

Further, we assume $\mathbb{E}[x_{i,d_1,p,q}] = \mathcal{X}_{d_1,q}$, which means the expectation of input feature only depends on output position $d_1$ and computational path $q$. Thus, Equation 25 is converted to:

$$
\begin{aligned}
\mathbb{E}[x_{j,d_1}^{(L)}] &= \frac{\rho}{\lambda^{(L-1)/2}} \sum_{v_i \in \mathbb{V}} \sum_{p=1}^{\psi} \sum_{q=1}^{\phi} \mathcal{X}_{d_1,q} \prod_{l=1}^{L} a_{i,p}^{(l)} W_{d_1,q}^{(l)} \\
&= \frac{\rho}{\lambda^{(L-1)/2}} \sum_{v_i \in \mathbb{V}} \left( \sum_{q=1}^{\phi} \mathcal{X}_{d_1,q} \prod_{l=1}^{L} W_{d_1,q}^{(l)} \right) \left( \sum_{p=1}^{\psi} \prod_{l=1}^{L} a_{i,p}^{(l)} \right) \\
&= \sum_{v_i \in \mathbb{V}} \left( \frac{\rho}{\lambda^{(L-1)/2}} \sum_{q=1}^{\phi} \mathcal{X}_{d_1,q} \prod_{l=1}^{L} W_{d_1,q}^{(l)} \right) P_L^{j \to i}. \\
&= \sum_{v_i \in \mathcal{N}(j)} \mathbb{E}[x_{d_1}^{'}] P_L^{j \to i}.
\end{aligned}
\tag{26}
$$

The final line is due to $\mathbb{E}[x_{d_1}^{'}] = \frac{\rho}{\lambda^{(L-1)/2}} \sum_{q=1}^{\phi} \mathcal{X}_{d_1,q} \prod_{l=1}^{L} W_{d_1,q}^{(l)}$, and $P_L^{j \to i} = 0$ for any node except neighborhoods of $v_j$, $\mathcal{N}(j)$. $\mathbb{E}[x_{d_1}^{'}]$ is the expectation of final output for node $v_i$ at dimension $d_1$ by a standard neural network, **without neighborhood information aggregation** as in GNNs. This means each node's features are processed independently through the layers.

By Definition 3.2 and Theorem 3.1, we estimate $x_{j,d_1}^{(L)}$ with $\hat{h}_{j,d_1}$, which can be calculated through random walk:

$$
\hat{h}_{j,d_1}^{(L)}(\mathbb{V}_r^{p\prime}) = \sum_{v_i \in \mathbb{V}_l \cup \mathbb{S} \cup \mathbb{V}_{r-1}^p} \hat{x}_{i,d_1}^{(L)} P_{\infty}^{j \to i},
\tag{27}
$$

where $\hat{x}_{i,d_1}^{(L)}$ enote the output logit by trained teacher model $f_{r-1}$, while $\hat{h}_{j,d_1}$ denotes our estimation for $\tilde{x}_{j,d_1}^{(L)}$, the true logit by the untrained student model $f_r$. As defined, compared with Equation 26, we only estimate with logits of **labeled or pseudo-labeled nodes** in Equation 27.

As all logits will go through Equation 16, which convert them to class probabilities by softmax function. By Lemma B.1, if the logit estimation error, $\delta_{j,d_1} = x_{j,d_1}^{(L)} - \hat{h}_{j,d_1}^{(L)}(\mathbb{V}_r^{p\prime})$, is small for all dimensions $d_1$, then the final probabilities computed from $\hat{h}_{j,d_1}^{(L)}(\mathbb{V}_r^{p\prime})$ are not far from ground-truth probabilities. Thus, the entropy estimation of node $v_j$ is still accurate.

### B.5.3 1 LAYER GNN

Now we consider a 1-layer GNN, such that $L = 1$. Recall Equation 17, the output for node $v_j$ by a standard neural network (NN) without aggregation can be represented as:

$$
x_{j,d_1}^{(1)} = \rho \sum_{q=1}^{\phi} x_{j,d_1,q} W_{d_1,q}^{(1)}.
\tag{28}
$$

Now, since only 1 layer is employed, we simply the notation to distinguish variables from $(r-1)$-th and $r$-th round. Denote $x_{i,d_1}^{r-1'}$ and $x_{i,d_1}^{r'}$ as the output logits by standard NN at $(r-1)$-th and $r$-th round, and similarly, $W_{d_1,q}^{r-1}$ for weights at $(r-1)$-th round and $W_{d_1,q}^{r-1}$ for $r$-th round. Since we assume initial feature $\mathcal{X}_{d_1,q}$ only depends on $q$ and $d_1$, and input feature $x_{d_1,q}$ is fixed, the expectation of output $x_{i,d_1}^{r-1'}$ is

$$
\mathbb{E}[x_{j,d_1}^{r-1'}] = \rho \sum_{q=1}^{\phi} \mathcal{X}_{d_1,q} W_{d_1,q}^{r-1},
\tag{29}
$$

where no right-hand item is related with node $j$. Similarly, the expectation of output $x_{i,d_1}^{r'}$ is

$$\mathbb{E}[x_{j,d_1}^{r'}] = \rho \sum_{q=1}^{\phi} \mathcal{X}_{d_1,q} W_{d_1,q}^r. \tag{30}$$

In Definition 3.3, the sampled nodes are in set $\mathbb{S}$. Denote $\eta_{j,d_1}(\mathbb{S})$ as the logit estimation error of node $v_j$ at $d_1$ using $\mathbb{S}$. Note that by softmax conversion, for any $C$ dimensional variable $X$, $H(X) = H(aX)$, where $a$ is a constant for every dimension. Let $a = \frac{1}{\sum_{v_i \in \mathbb{V}_l \cup \mathbb{V}_{r-1}^P} P_\infty^{j \to i}}$. Then expanded by Equation 26, the logit estimation error of node $v_j$ at dimension $d_1$:

$$\begin{aligned}
\mathbb{E}[\delta_{j,d_1}(\mathbb{S})] &= \mathbb{E}[x_{j,d_1}^{(1)} - a\hat{h}_{j,d_1}^{(1)}(\mathbb{S})] \\
&= \mathbb{E}[x_{j,d_1}^{(1)}] - a \sum_{v_i \in \mathbb{V}_l \cup \mathbb{V}_{r-1}^P \cup \mathbb{S}} \mathbb{E}[\hat{x}_{i,d_1}^{(1)}] P_\infty^{j \to i} \\
&= \sum_{v_i \in \mathcal{N}(j)} \mathbb{E}[x_{i,d_1}^{r'}] P_1^{j \to i} - a \sum_{v_i \in \mathbb{V}_l \cup \mathbb{V}_{r-1}^P \cup \mathbb{S}} P_\infty^{j \to i} \sum_{v_k \in \mathcal{N}(i)} \mathbb{E}[x_{k,d_1}^{r-1'}] P_1^{k \to j} \\
&= \mathbb{E}[x_{i,d_1}^{r'}] - \frac{\sum_{v_i \in \mathbb{V}_l \cup \mathbb{V}_{r-1}^P \cup \mathbb{S}} P_\infty^{j \to i}}{\sum_{v_i \in \mathbb{V}_l \cup \mathbb{V}_{r-1}^P} P_\infty^{j \to i}} \mathbb{E}[x_{k,d_1}^{r-1'}].
\end{aligned} \tag{31}$$

The first to third line is expansion by prior definitions. The fourth line is according to Equation 29, that feature expectation only depends on output dimension $d_1$, and $\sum_{v_i \in \mathcal{N}(j)} P_1^{j \to i} = \sum_{v_k \in \mathcal{N}(i)} P_1^{k \to j} = 1$. Assume that activation paths remain the same in both round,

$$\begin{aligned}
\mathbb{E}[\delta_{j,d_1}(\mathbb{S})] &= \rho\Big(\sum_{q=1}^{\phi} \mathcal{X}_{d_1,q} W_{d_1,q}^{r-1} - \frac{\sum_{v_i \in \mathbb{V}_l \cup \mathbb{V}_{r-1}^P \cup \mathbb{S}} P_\infty^{j \to i}}{\sum_{v_i \in \mathbb{V}_l \cup \mathbb{V}_{r-1}^P} P_\infty^{j \to i}} \sum_{q=1}^{\phi} \mathcal{X}_{d_1,q} W_{d_1,q}^r\Big) \\
&= \rho \sum_{q=1}^{\phi} \mathcal{X}_{d_1,q}\Big(W_{d_1,q}^{r-1} - \frac{\sum_{v_i \in \mathbb{V}_l \cup \mathbb{V}_{r-1}^P \cup \mathbb{S}} P_\infty^{j \to i}}{\sum_{v_i \in \mathbb{V}_l \cup \mathbb{V}_{r-1}^P} P_\infty^{j \to i}} W_{d_1,q}^r\Big).
\end{aligned} \tag{32}$$

Though we can initialize network weights as the same, the loss function is computed differently, resulting difference in $W_{d_1,q}^{r-1}$ and $W_{d_1,q}^r$. Therefore, we need to further discuss the weight update equation. The loss function, e.g., cross-entropy loss, is set as the sum of prediction error on each labeled sample. That is, for $n$ samples, we have the loss at dimension $d_1$:

$$L_{d_1} = \sum_{v_i} L_{i,d_1}(y_{i,d_1}, \hat{y}_{i,d_1}), \hat{y}_{i,d_1} = Softmax(x_{i,d_1}^{'}), \tag{33}$$

where $L_{d_1}^{r-1}$ is the loss at $d_1$ position.

Therefore, consider the classic gradient descent with one step $\eta$, we can update initial weight $W_{d_1,q}$ in $(r-1)$-th round through:

$$W_{d_1,q}^{r-1} = W_{d_1,q} - \eta \frac{\partial L_{d_1}^{r-1}}{\partial W_{d_1,q}},$$

$$\frac{\partial L_{d_1}^{r-1}}{\partial W_{d_1,q}} = \sum_{v_i \in \mathbb{V}_l \cup \mathbb{V}_{r-1}^p} \frac{\partial L_{i,d_1}^{r-1}}{\partial W_{d_1,q}} \text{(Eq. 33)}$$

$$= \sum_{v_i \in \mathbb{V}_l \cup \mathbb{V}_{r-1}^p} \frac{\partial L_{i,d_1}^{r-1'}}{\partial x_{i,d_1}^{r-1'}} \frac{\partial x_{i,d_1}^{r-1'}}{\partial W_{d_1,q}} \text{ (chain rule)}$$

$$= \rho \sum_{v_i \in \mathbb{V}_l \cup \mathbb{V}_{r-1}^p} \frac{\partial L_{i,d_1}}{\partial x_{i,d_1}'} \sum_{q=1}^{\phi} x_{i,d_1,q} \text{ (Eq. 28)}$$

$$(34)$$

Similarly, we derive weight update for $r$-th round:

$$W_{d_1,q}^r = W_{d_1,q} - \eta \frac{\partial L_{d_1}^r}{\partial W_{d_1,q}},$$

$$\frac{\partial L_{d_1}^r}{\partial W_{d_1,q}} = \sum_{v_i \in \mathbb{V}_l \cup \mathbb{V}_{r-1}^p \cup \mathbb{S}} \frac{\partial L_{i,d_1}^r}{\partial W_{d_1,q}}$$

$$= \rho \sum_{v_i \in \mathbb{V}_l \cup \mathbb{V}_{r-1}^p \cup \mathbb{S}} \frac{\partial L_{i,d_1}}{\partial x_{i,d_1}'} \sum_{q=1}^{\phi} x_{i,d_1,q}.$$

$$(35)$$

Note that as both equations are initialized with the same weight matrix $\boldsymbol{W}$ and activation path, according to Equation 28, $x_{i,d_1}^{r-1'} = x_{i,d_1}^{r'}$. Further, since labels are fixed for $\mathbb{V}_l \cup \mathbb{V}_{r-1}^p$, so $\frac{\partial L_{i,d_1}}{\partial x_{i,d_1}}$ is also the same. Therefore, in the final line of both equations, we remove $r$ or $r-1$ from the superscripts. Substituting Equation 34 and 35 into 32, we get:

$$\mathbb{E}[\delta_{j,d_1}(\mathbb{S})] = \rho \sum_{q=1}^{\phi} \mathcal{X}_{d_1,q} [(W_{d_1,q} - \eta \frac{\partial L_{d_1}^{r-1}}{\partial W_{d_1,q}}) - \frac{\sum_{v_i \in \mathbb{V}_l \cup \mathbb{V}_{r-1}^p \cup \mathbb{S}} P_\infty^{j \to i}}{\sum_{v_i \in \mathbb{V}_l \cup \mathbb{V}_{r-1}^p} P_\infty^{j \to i}} (W_{d_1,q} - \eta \frac{\partial L_{d_1}^r}{\partial W_{d_1,q}})]$$

$$= \rho \sum_{q=1}^{\phi} \mathcal{X}_{d_1,q} [(1 - \frac{\sum_{v_i \in \mathbb{V}_l \cup \mathbb{V}_{r-1}^p \cup \mathbb{S}} P_\infty^{j \to i}}{\sum_{v_i \in \mathbb{V}_l \cup \mathbb{V}_{r-1}^p} P_\infty^{j \to i}}) W_{d_1,q}$$

$$- \eta \rho (\sum_{v_i \in \mathbb{V}_l \cup \mathbb{V}_{r-1}^p} \frac{\partial L_{i,d_1}}{\partial x_{i,d_1}'} \sum_{q=1}^{\phi} \mathcal{X}_{d_1,q}$$

$$- \frac{\sum_{v_i \in \mathbb{V}_l \cup \mathbb{V}_{r-1}^p \cup \mathbb{S}} P_\infty^{j \to i}}{\sum_{v_i \in \mathbb{V}_l \cup \mathbb{V}_{r-1}^p} P_\infty^{j \to i}} \sum_{v_i \in \mathbb{V}_l \cup \mathbb{V}_{r-1}^p \cup \mathbb{S}} \frac{\partial L_{i,d_1}}{\partial x_{i,d_1}'} \sum_{q=1}^{\phi} \mathcal{X}_{d_1,q})].$$

$$(36)$$

According to Lemma B.2, when $\delta_{j,d_1}(\mathbb{S}) - \delta_{j,d_2}(\mathbb{S})$ is small for any $d_1$ and $d_2$, the final entropy estimation is bounded.

$$\frac{1}{\rho}\mathbb{E}[\delta_{j,d_1}(\mathbb{S}) - \delta_{j,d_2}(\mathbb{S})]$$

$$=\frac{1}{\rho}(\mathbb{E}[x^{r'}_{j,d_1} - x^{r'}_{j,d_2}] - \frac{\sum_{v_i \in \mathbb{V}_l \cup \mathbb{V}^p_{r-1} \cup \mathbb{S}} P^{j \to i}_\infty}{\sum_{v_i \in \mathbb{V}_l \cup \mathbb{V}^p_{r-1}} P^{j \to i}_\infty} \mathbb{E}[x^{r-1'}_{j,d_1} - x^{r-1'}_{j,d_2}])$$

$$=\sum_{q=1}^{\phi}(\mathcal{X}_{d_1,q}W^r_{d_1,q} - \mathcal{X}_{d_2,q}W^r_{d_2,q}) - \frac{\sum_{v_i \in \mathbb{V}_l \cup \mathbb{V}^p_{r-1} \cup \mathbb{S}} P^{j \to i}_\infty}{\sum_{v_i \in \mathbb{V}_l \cup \mathbb{V}^p_{r-1}} P^{j \to i}_\infty}(\mathcal{X}_{d_1,q}W^{r-1}_{d_1,q} - \mathcal{X}_{d_2,q}W^{r-1}_{d_2,q})]$$

$$=[\sum_{q=1}^{\phi}\mathcal{X}_{d_1,q}(W_{d_1,q} - \eta \sum_{v_i \in \mathbb{V}_l \cup \mathbb{V}^p_{r-1} \cup \mathbb{S}} \frac{\partial L_{i,d_1}}{\partial W_{d_1,q}}) \tag{37}$$

$$-\mathcal{X}_{d_2,q}(W_{d_2,q} - \eta \sum_{v_i \in \mathbb{V}_l \cup \mathbb{V}^p_{r-1} \cup \mathbb{S}} \frac{\partial L_{i,d_2}}{\partial W_{d_2,q}})]$$

$$-\frac{\sum_{v_i \in \mathbb{V}_l \cup \mathbb{V}^p_{r-1} \cup \mathbb{S}} P^{j \to i}_\infty}{\sum_{v_i \in \mathbb{V}_l \cup \mathbb{V}^p_{r-1}} P^{j \to i}_\infty}[\sum_{q=1}^{\phi}\mathcal{X}_{d_1,q}(W_{d_1,q} - \eta \sum_{v_i \in \mathbb{V}_l \cup \mathbb{V}^p_{r-1}} \frac{\partial L_{i,d_1}}{\partial W_{d_1,q}})$$

$$-\mathcal{X}_{d_2,q}(W_{d_2,q} - \eta \sum_{v_i \in \mathbb{V}_l \cup \mathbb{V}^p_{r-1}} \frac{\partial L_{i,d_2}}{\partial W_{d_2,q}})].$$

This is to say, the entropy estimation is accurate when the predicted output logits difference between $d_1$ and $d_2$ by student model $f_r$, $\mathbb{E}[x^{r'}_{j,d_1} - x^{r'}_{j,d_2}]$, is in proportion to the difference by teacher model $f_{r-1}$, $\mathbb{E}[x^{r-1'}_{j,d_1} - x^{r-1'}_{j,d_2}]$, i.e., $\mathbb{E}[\frac{x^{r'}_{j,d_1} - x^{r'}_{j,d_2}}{x^{r-1'}_{j,d_1} - x^{r-1'}_{j,d_2}}] = \frac{\sum_{v_i \in \mathbb{V}_l \cup \mathbb{V}^p_{r-1} \cup \mathbb{S}} P^{j \to i}_\infty}{\sum_{v_i \in \mathbb{V}_l \cup \mathbb{V}^p_{r-1}} P^{j \to i}_\infty}$. This is aligned with the gradient of $W^r$ and $W^{r-1}$: $\mathbb{V}_l \cup \mathbb{V}^p_{r-1} \cup \mathbb{S}$ and $\mathbb{V}_l \cup \mathbb{V}^p_{r-1}$.

### B.6 Robustness of $k$-Bounded Banzhaf Value

In this section, we recall conditions in Theorem 3.2 – the distinguishability of ground-truth utility between $v_i$ and $v_j$ on coalitions less than size $k$ is large enough, $\min_{m \le k-1} \Delta^{(m)}_{i,j}(U) \ge \tau$, and the utility perturbation is small such that $\left\|\hat{U} - U\right\|_2 \le \tau \sqrt{\sum_{m=1}^{k-1} \binom{|\mathbb{V}^u_{r-1}|-2}{m-1}}$.

*Proof.* We begin by recalling the definition of $\Delta^{(m)}_{i,j}(U)$ and $D_{i,j}(U)$ as:

$$\Delta^{(m)}_{i,j}(U) = \binom{|\mathbb{V}^u_{r-1}| - 2}{m - 1}^{-1} \sum_{\mathbb{S} \subseteq \mathbb{V}^u_{r-1} \setminus \{v_i, v_j\}, |\mathbb{S}| = m-1}[U(\mathbb{S} \cup v_i) - U(\mathbb{S} \cup v_j)], \tag{38}$$

and

$$D_{i,j}(U) = \frac{k}{n_s} \sum_{m=1}^{k-1} \binom{|\mathbb{V}^u_{r-1}| - 2}{m - 1} \Delta^{(m)}_{i,j}(U). \tag{39}$$

Then, we rewrite $D_{i,j}(\hat{U})$ as a dot product of $U$ and a column vector $\boldsymbol{a} \in \mathbb{R}^{n_s}$:

$$D_{i,j}(\hat{U}) = \boldsymbol{a}^T U, \tag{40}$$

where each entry of $\boldsymbol{a}$ corresponds to a subset $\mathbb{S}$. Then,

$$D_{i,j}(U)D_{i,j}(\hat{U}) = (\boldsymbol{a}^T U)(\boldsymbol{a}^T \hat{U}) = (\boldsymbol{a}^T U)^T(\boldsymbol{a}^T \hat{U}) = U^T \boldsymbol{a}\boldsymbol{a}^T \hat{U} = U^T \boldsymbol{A} \hat{U}. \tag{41}$$

Thus, since $AA = (aa^T)(aa^T) = a(a^Ta)a^T = (a^Ta)aa^T = (a^Ta)A$, then

$$\frac{|U^T AU|}{\|U^T A\|_2} = \frac{|U^T AU|}{\sqrt{U^T AAU}} = \frac{|U^T AU|}{\sqrt{a^Ta}\sqrt{|U^T AU|}} = \sqrt{\frac{|U^T AU|}{a^Ta}}. \tag{42}$$

As the weight of Banzhaf value is the same for all subset, and by assumption that $\min_{m \leq k-1} \Delta_{i,j}^{(m)}(U) \geq \tau$, we can rewrite $\frac{|U^T AU|}{a^Ta}$ as

$$\frac{|\sum_{\mathbb{S}_1 \subset \mathbb{V}_{r-1}^u \backslash \{v_i, v_j\}}^{|\mathbb{S}_1| \leq k-2} \sum_{\mathbb{S}_2 \subset \mathbb{V}_{r-1}^u \backslash \{v_i, v_j\}}^{|\mathbb{S}_2| \leq k-2} (\frac{k}{n_s})^2 (U(\mathbb{S}_1 \cup v_i) - U(\mathbb{S}_1 \cup v_j))(U(\mathbb{S}_2 \cup v_i) - U(\mathbb{S}_2 \cup v_j))|}{\sum_{\mathbb{S} \subset \mathbb{V}_{r-1}^u \backslash \{v_i, v_j\}}^{|\mathbb{S}| \leq k-2} (\frac{k}{n_s})^2}$$

$$= \frac{(\sum_{\mathbb{S} \subset \mathbb{V}_{r-1}^u \backslash \{v_i, v_j\}}^{|\mathbb{S}| \leq k-2} \frac{k}{n_s} (U(\mathbb{S} \cup v_i) - U(\mathbb{S} \cup v_j))^2}{\sum_{\mathbb{S} \subset \mathbb{V}_{r-1}^u \backslash \{v_i, v_j\}}^{|\mathbb{S}| \leq k-2} (\frac{k}{n_s})^2}$$

$$= \frac{(\sum_{m=1}^{k-1} \binom{|\mathbb{V}_{r-1}^u|-2}{m-1} \frac{k}{n_s} \binom{|\mathbb{V}_{r-1}^u|-2}{m-1}^{-1} \sum_{\mathbb{S} \subset \mathbb{V}_{r-1}^u \backslash \{v_i, v_j\}}^{|\mathbb{S}| = m-1} \frac{k}{n_s} (U(\mathbb{S} \cup v_i) - U(\mathbb{S} \cup v_j)))^2}{\sum_{m=1}^{k-1} \binom{|\mathbb{V}_{r-1}^u|-2}{m-1} (\frac{k}{n_s})^2}$$

$$= \frac{(\sum_{m=1}^{k-1} \binom{|\mathbb{V}_{r-1}^u|-2}{m-1} \frac{k}{n_s} \Delta_{i,j}^{(m)}(U))^2}{\sum_{m=1}^{k-1} \binom{|\mathbb{V}_{r-1}^u|-2}{m-1} (\frac{k}{n_s})^2}$$

$$\geq \frac{(\sum_{m=1}^{k-1} \binom{|\mathbb{V}_{r-1}^u|-2}{m-1} \frac{k}{n_s})^2 \tau^2}{\sum_{m=1}^{k-1} \binom{|\mathbb{V}_{r-1}^u|-2}{m-1} (\frac{k}{n_s})^2}$$

$$= \sum_{m=1}^{k-1} \binom{|\mathbb{V}_{r-1}^u| - 2}{m - 1} \tau^2. \tag{43}$$

Therefore, $\sqrt{\frac{|U^T AU|}{a^Ta}} \geq \sqrt{\sum_{m=1}^{k-1} \binom{|\mathbb{V}_{r-1}^u|-2}{m-1}} \tau$. Further, since the utility perturbation is small such that $\left\|\hat{U} - U\right\|_2 \leq \tau \sqrt{\sum_{m=1}^{k-1} \binom{|\mathbb{V}_{r-1}^u|-2}{m-1}}$, and by Equation 42, then $\left\|\hat{U} - U\right\|_2 \leq \sqrt{\frac{|U^T AU|}{a^Ta}} = \frac{|U^T AU|}{\|U^T A\|_2}$, which equals $|U^T A(\hat{U} - U)| \leq \left\|U^T A\right\|_2 \left\|\hat{U} - U\right\|_2 \leq |U^T AU|$ by triangle inequality. Finally, with Equation 41,

$$D_{i,j}(U)D_{i,j}(\hat{U}) = (\hat{\phi}(i) - \hat{\phi}(j))(\phi(i) - \phi(j)) = U^T A\hat{U} = U^T A((\hat{U} - U) + U) \geq 0. \tag{44}$$

$\square$

## C ALGORITHM FORMULATION

In this section, we provide pseudo-code in Algorithm 1 and pipeline figure in Figure 3 for our method, BANGS.

### C.1 TIME COMPLEXITY

- Line 1: Add self-loop takes $O(|\mathbb{V}|)$, and row normalization takes $O(|\mathbb{V}|^2)$.
- Line 2: This depends on model selection. Consider a simple GNN model with number of layers $L$ and hidden units $n_u$, and train $n_e$ epochs. The time complexity is $O(Ln_u n_e |\mathbb{V}|^2)$.
- Line 5: Softmax activation takes $O(C|\mathbb{V}|)$, $C$ is the number of classes.

---

**Algorithm 1** BANGS

---

**Input:** Graph $\mathcal{G}$ including $A$, $X$ and $y^l$, Iteration Number $R$, Candidate Node Number $K$, Selection Node Number $k$, Sample Size $k$ and Number $B$, Confidence Calibration Model $f_c$

1: Calculate $\hat{A}$ by $A$, with added self-loop and row normalization.
2: Train the initial teacher model $f_0$.
3: **for** Every iteration $r \in [1, ...R]$ **do**
4:      Employ $f_{r-1}$ as teacher model.
5:      Calculate confidence for all nodes $V$ with maximum softmax output of $f_{r-1}$.
6:      Calibrate confidence with $f_c$.
7:      Select the top $K$ confident nodes as candidate node set $S_K \subset V$.
8:      Store the initial logits propagated by current labeled nodes $\mathbb{V}^l \cup \mathbb{V}^p_{r-1}$.
9:      Sample $B$ times candidate set $\mathbb{S} \subset \mathbb{V}^u_{r-1}$, $|\mathbb{S}| \le k$
10:      **for** Every $\mathbb{S}$ **do**
11:         Add initial logits by the new logits propagated by $\mathbb{S}$.
12:         Get entropy and calculate utility function $U(\mathbb{S})$ by Eq. 2.
13:      **end for**
14:      Calculate Banzhaf value $\phi(i; U, \mathbb{V}^u_{r-1})$ for every $v_i$ accordingly.
15:      Select top $k$ nodes of highest Banzhaf value from $K$ candidate nodes to label $\mathbb{V}^p_r$, and get $\hat{y}^p_r$, the predicted label of $\mathbb{V}^p_r$ by $f_{r-1}$,
16:      Train student model $f_r$ on $Y^l$ and $\hat{y}^p_r$.
17: **end for**

---

- Line 6: The calibration process also depends the selection of claibration model. Consider Temperature Scaling with $n_{TS}$ iterations. This takes $O(n_{TS}C|\mathbb{V}|)$.

- Line 7: Top-$K$ node takes $O(|\mathbb{V}^u_{r-1}|log(|\mathbb{V}^u_{r-1}|))$.

- Line 8: This step only propagates from $\mathbb{V}\backslash\mathbb{V}^u_{r-1} = \mathbb{V}^l \cup \mathbb{V}^p_{r-1}$. Consider propagate $H$ times. So it takes $O(H|\mathbb{V}||\mathbb{V}\backslash\mathbb{V}^u_{r-1}|C)$.

- Line 9: Sampling takes $O(B|\mathbb{V}^u_{r-1}|)$.

- Line 10 -13: Similarly, line 11 takes $O(H|\mathbb{V}|kC)$, and line 12 takes $O(kC)$. For all samples, the total running time is $O(B|\mathbb{V}|kC)$.

- Line 14: Using MSR, this could be done in a linear time to sample num $- O(B)$.

- Line 15: The Top-$k$ ranking is no longer than Top-$K$, as $K \ge k$.

- Line 16: Same as line 2, takes $O(Ln_un_e|\mathbb{V}|^2)$.

In summary, the initialization takes $O(Ln_un_e|\mathbb{V}|^2)$. In each round, 1) confidence estimation takes $O(n_{TS}C|\mathbb{V}|)$; 2) node selection takes $O(H|\mathbb{V}|(|\mathbb{V}\backslash\mathbb{V}^u_{r-1}| + Bk)C)$; 3) retraining also takes $O(Ln_un_e|\mathbb{V}|^2)$. Given $R$ self-training rounds, the total complexity is roughly $O(R|\mathbb{V}|(H(|\mathbb{V}| + Bk)C + Ln_un_e|\mathbb{V}|))$. The final running time mainly depends on the selection of GNN structure, alongwith Banzhaf sampling number and size.

## C.2 SPACE COMPLEXITY

We have provided illustration in Sec. 3.3. The bottleneck is in loading graph and utility function computation. We can control space complexity to $O(|E^b||V^b|C)$ at the $r$-th iteration, where $V^b$ means the batch of labeled nodes, and $E^b$ denotes edges connected to $V^b$ in $\hat{A}$. Nonetheless, the tradeoff between space and time complexity needs consideration. For smaller graphs whose influence matrix can be stored, it takes shorter time to first compute influence matrix for all nodes, and query values during node selection.

## D IMPLEMENTATION DETAILS

### D.1 DATASETS AND CODE

The statistics of datsets are listed in Table D.1. Datasets used in this paper could be found in:

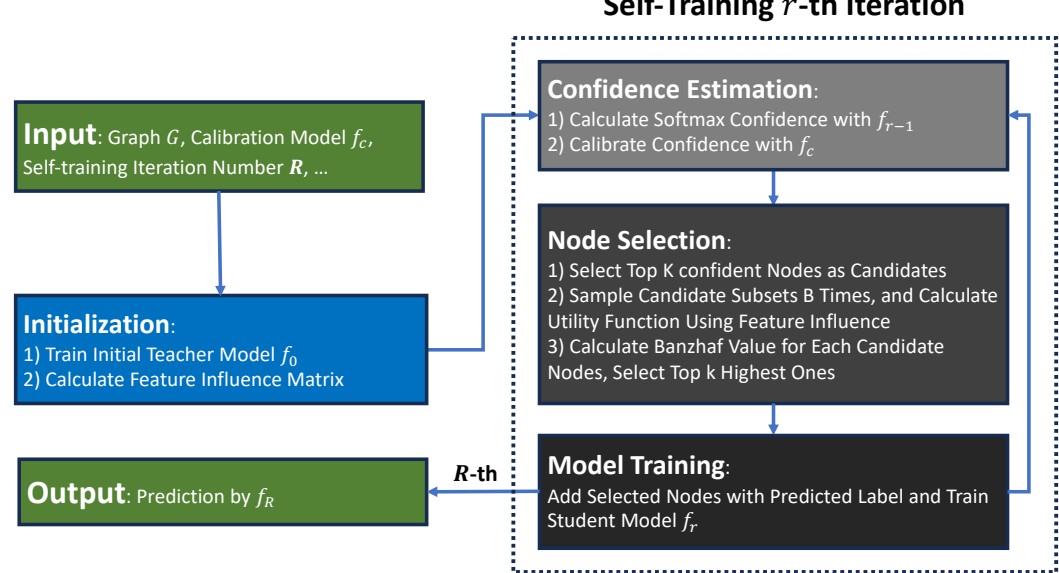

Figure 3: Pipeline of BANGS

- Cora, Citeseer and PubMed (Yang et al., 2016) (`https://github.com/kimiyoung/planetoid`);

- LastFM (Rozemberczki & Sarkar, 2020) (`https://github.com/benedekrozemberczki/FEATHER`).

- Flickr (Zeng et al., 2019) (`https://github.com/GraphSAINT/GraphSAINT`).

We employ the re-packaged datasets from PyG (Fey & Lenssen, 2019) (`https://github.com/pyg-team/pytorch_geometric`, version 2.5.2).

Baseline methods:

- M3S (Sun et al., 2020) (`https://github.com/datake/M3S`). We use the reproduced version in CPL;

- CaGCN (Wang et al., 2021) (`https://github.com/BUPT-GAMMA/CaGCN`);

- GATS (Hsu et al., 2022) (`https://github.com/hans66hsu/GATS`);

- DR-GST (Liu et al., 2022a) (`https://github.com/BUPT-GAMMA/DR-GST`). We use the reproduced version in CPL;

- CPL (Botao et al., 2023) (`https://github.com/AcEbt/CPL`)

All code and datasets used in this paper, if presented, are open-sourced under MIT license.

Table 5: Dataset statistics.

| Dataset | Cora | Citeseer | PubMed | LastFMAsia | Flickr |
|---|---|---|---|---|---|
| **#Nodes** | 2,708 | 3,317 | 19,171 | 7,624 | 89,250 |
| **#Links** | 10,556 | 9,104 | 88,648 | 5,5612 | 899,756 |
| **#Features** | 1,433 | 3,703 | 500 | 128 | 500 |
| **#Classes** | 7 | 6 | 3 | 18 | 7 |

Our own codes are provided in `https://github.com/fangxin-wang/BANGS`.

## D.2 COMPUTING RESOURCES

The experiments are mainly running in a machine with NVIDIA GeForce GTX 4090 Ti GPU with 24 GB memory, and 80 GB main memory. Some experiments of small graphs are conducted on a MacBook Pro with Apple M1 Pro Chip with 16 GB memory.

## E  EXPERIMENT DETAILS

### E.1  ADDITIONAL EXPERIMENT RESULTS

In this section, we include and analyze additional experiment results.

Table 6: Node classification accuracy (%) with graph self-training strategies on different datasets. Baseline methods report final test accuracy, our method reports test accuracy by early stopping.

| Base Model | Baselines | | | | | | BANGS |
|---|---|---|---|---|---|---|---|
| | Raw | M3S | CaGCN | DR-GST | Random | CPL | |
| Cora | 80.82±0.14 | 81.38±0.75 | 83.06±0.11 | 82.03±0.93 | 82.43±0.21 | 83.00±0.57 | **83.47±0.54*** |
| Citeseer | 70.18±0.27 | 68.80±0.53 | 72.84±0.07 | 71.10±0.88 | 72.50±0.52 | 72.45±0.54 | **73.23±0.70**** |
| PubMed | 78.40±0.11 | 79.10±0.28 | **81.16±0.10** | 77.42±0.60 | 78.93±0.47 | 79.37±0.63 | 81.03±0.56 |
| LastFM | 78.07±0.31 | 78.32±0.81 | 79.60±1.02 | 79.92±0.44 | 79.13±0.11 | 80.56±1.22 | **82.66±0.43**** |

In Table 6, we include test accuracy that is early stopped by validation accuracy. Our methods are the best or second best across all datasets. Nonetheless, it is worth noticing that though CPL tends to achieve best test accuracy in the self-training iteration, CaGCN tends to perform better with early-stopping criterion. This is because the confidence of test data are calibrated using validation data in CaGCN, such that test accuracy is high using with validation accuracy.

Table 7: Total running time (seconds) comparison for BANGS and CPL on datasets with GCN or GAT as base models.

| Dataset | BANGS (GCN) | CPL (GCN) | BANGS (GAT) | CPL (GAT) |
|---|---|---|---|---|
| Cora | 201.82 | 118.64 | 288.54 | 326.62 |
| Citeseer | 242.40 | 184.48 | 422.88 | 432.39 |
| Pubmed | 423.40 | 320.22 | 954.87 | 929.13 |
| LastFM | 325.62 | 209.80 | 725.54 | 697.88 |

Table 8: Iteration block average running time comparison (s) on Cora.

| Base Model | Method | Sample Size | Confidence Estimation | Node Selection | Model Training |
|---|---|---|---|---|---|
| GCN | CPL | NA | 0.04 | 0.00 | 3.45 |
| GCN | Ours | 100 | 2.45 | 0.80 | 4.08 |
| GCN | Ours | 200 | 2.41 | 1.33 | 3.69 |
| GAT | CPL | NA | 0.06 | 0.00 | 11.20 |
| GAT | Ours | 100 | 0.16 | 0.80 | 10.20 |
| GAT | Ours | 200 | 0.16 | 1.31 | 10.76 |

In Table 7, we use GCN with simple network structure, our methods are slower than CPL; while using GAT that takes longer inference time, our methods is comparable to CPL. In Table 8, we further show the running times in one seed to compare with the CPL.

In Table 9, we compare running time and test performance of our method with the raw model and best baseline, CPL, on the same setting as described in Section 4. We test on Flickr dataset with ∼90k nodes and the additional obgn-arxiv (Hu et al., 2020) dataset with ∼170k nodes. We observe that graph self-training methods often exhibit comparatively smaller improvements on larger datasets. To address this problem, additional techniques, such as contrastive sampling (Zhou et al., 2023), may be incorporated as a future direction. The total running time of our algorithm also

Table 9: Running time and accuracy comparison on larger datasets.

| Dataset | Methods | Running Time (s) | Accuracy (%) |
|---|---|---|---|
| **Flickr** | GCN | 15.16 | 49.53±0.11 |
| | CPL | 633.35 | 50.02±0.22 |
| | BANGS | 1189.48 | **50.23±0.25** |
| **obgn-arxiv** | GCN | 30.81 | 63.82±0.12 |
| | CPL | 1287.57 | 63.81±0.17 |
| | BANGS | 1668.98 | **63.86±0.18** |

exhibits increase in larger graphs, largerly due to computation involves adjacency matrix in PPR calculation. In this case, mini-batching PPR (Gasteiger et al., 2022) is a good choice.

### E.2    CASE STUDY: CONFIDENCE CALIBRATION

To motivate the integration of calibration, we perform a case study. In Figure 4, we predict labels of PubMed data, which have only 3 classes and exhibits high confidence. The other hyperparameters except node selection criterion are fixed. To select informative nodes from highly confident ones, our framework sets the top $K = 200$ confident nodes as candidates and selects $k = 100$ nodes with the highest Banzhaf value (Definition 3.3). Selecting nodes with the highest confidence, denoted by Conf (Uncal), would bring slightly better performances compared with the raw GCN predictions. Surprisingly, calibrating the same with CaGCN (Wang et al., 2021), denoted by Conf (cal), produces worse performance under raw GCN. In contrast, with uncalibrated confidence, our framework produces comparable or better results than Conf (Uncal). However, with calibrated confidence, our framework exhibits a significant improvement. In the experiments (Section 4.3), we observe similar trends for most datasets.

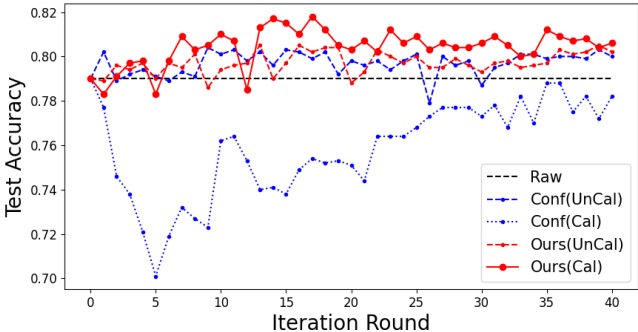

Figure 4: The test accuracy of PubMed data with different node selection criteria in self-training iteration with 40 rounds. In each round, 100 nodes are selected to pseudo-label. Our method with confidence calibration outperforms others.

