# OpenReview forum: "BANGS: Game-theoretic Node Selection for Graph Self-Training"
_ICLR.cc/2025/Conference — ICLR 2025 Poster_

### Official Review · Reviewer_oNSD · 2024-11-01

**Soundness:** 3
**Presentation:** 3
**Contribution:** 2
**Rating:** 6
**Confidence:** 4

**Summary:**

This paper presents a new node selection method for graph self-training. The authors leverage PPR-based feature propagation to estimate a utility function aimed at information gain. To address node interdependencies in graph-structured data, the authors employ Banzhaf values, allowing for combinatorial modeling of node influence during the node selection phase.

**Strengths:**

- The paper's motivation is clear and well-presented.
- The introduction of Banzhaf values to incorporate node interdependencies is novel and unique.
- BANGS is grounded in a profound theoretical foundation, which makes the proposed method more solid.

**Weaknesses:**

**W1.** BANGS requires a confidence calibration model, which further increases the computational costs. Additionally, the authors do not provide a running time analysis on the Flickr, a large-scale dataset; the results are selectively reported on small- or medium-sized datasets. Although the method is theoretically well-founded, its practical utility seems to be limited due to the high computational burden.

**W2.** The most concerning part is the marginal empirical performance. This is particularly noticeable in the ablative study, where the performance gap between Conf (Cal) and BANGS (Uncal) is minimal. Since the calibration model is not this paper's contribution, it is difficult to acknowledge the method's distinct efficacy. This calls into question about whether the high computational complexity of BANGS justifies the gains. Furthermore, the use of calibration model is somewhat hidden from the reader, only becoming clear in the experimental setting paragraph. For a paper that emphasizes empirical performance, it is critical to clearly identify the impactful components in the main text.

**W3.** The authors’ claim that "previous approaches all suffer from independent selection" (in introduction) may be overstated. For instance, CaGCN [1] propagates node confidence across the graph structure, reflecting combinatorial interactions through multi-hop neighbor consideration during calibration.

---
**References**

[1] Be Confident! Towards Trustworthy Graph Neural Networks via Confidence Calibration, NeurIPS 2021

[2] What Makes Graph Neural Networks Miscalibrated?, NeurIPS 2022

[3] Graph random neural networks for semi-supervised learning on graphs, NeurIPS 2020

**Questions:**

**Q1.** CaGCN is not considered a SOTA calibration model. Could the authors compare the performance between BANGS and GATS [2]? While GATS does not include a self-training experiment, it could likely be adapted to a pseudo-labeling framework, similar to CaGCN.

**Q2.** Comparison with consistency regularization approaches, such as [3], is highly recommended, as the current baselines are all pseudo-labeling methods.

---

> ### Author Response · Authors · 2024-11-20
>
> Thank you for your detailed reviews and suggested experiments. We have addressed them below.
>
> > W1. BANGS requires a confidence calibration model, which further increases the computational costs. Additionally, the authors do not provide a running time analysis on the Flickr, a large-scale dataset; the results are selectively reported on small- or medium-sized datasets. Although the method is theoretically well-founded, its practical utility seems to be limited due to the high computational burden.
>
>
> **Response:** Thanks for the suggestions. In the table below, we compare running time and test performance of our method with the raw model and best baseline, CPL, on the same setting as described in Section 4.1. The setting are the same as described in the main paper. We test on Flickr dataset with ~90k nodes and obgn-arxiv [a] dataset with ~170k nodes. We recognize that graph self-training methods often exhibit comparatively smaller improvements on larger datasets. To address this problem, additional techniques, such as contrastive sampling [b], may be incorporated as a future direction.
>
> Table 1. Running Time and Accuracy Comparsion of Flickr and obgn-arxiv Dataset.
> |  Dataset   |  Methods|Running Time(s)  | Accuracy (%)
> | --------| -------- | ------- | ----|
> | Flickr |  GCN | 15.16 | 49.53+-0.11|
> | Flickr |  CPL | 633.35 | 50.02+-0.22|
> | Flickr |  BANGS | 1189.48 | **50.23+-0.25** |
> | obgn-arxiv |  GCN | 30.81 | 63.82+-0.12 |
> | obgn-arxiv| CPL | 1287.57 | 63.81+-0.17 |
> | obgn-arxiv| BANGS | 1668.98 |  **63.86+-0.18** |
>
>
>
> [a] Wang, K., Shen, Z., Huang, C., Wu, C. H., Dong, Y., & Kanakia, A. (2020). Microsoft academic graph: When experts are not enough. Quantitative Science Studies, 1(1), 396-413.
>
> [b] Zhou, Z., Shi, J., Zhang, S., Huang, Z., & Li, Q. (2023). Effective stabilized self-training on few-labeled graph data. Information Sciences, 631, 369-384.
>
>
> > W2. The most concerning part is the marginal empirical performance. This is particularly noticeable in the ablative study, where the performance gap between Conf (Cal) and BANGS (Uncal) is minimal. Since the calibration model is not this paper's contribution, it is difficult to acknowledge the method's distinct efficacy. This calls into question about whether the high computational complexity of BANGS justifies the gains. Furthermore, the use of calibration model is somewhat hidden from the reader, only becoming clear in the experimental setting paragraph. For a paper that emphasizes empirical performance, it is critical to clearly identify the impactful components in the main text.
>
>
> **Response:** We understand your concern. Admittedly, confidence calibration is an important componement that could further enhance the performance of framework. We put it into a section in experiment part for the following reasons: 1) first, we want to keep a clear storyline in the methodology part -- caliration methods are not the main contribution of our work; 2) moreover, we think graph calibration is a general and foundamental problem in graph learning, due to unmatched prediction probability estimation.
>
> We argue that the close performance of Conf (Cal) and BANGS (Uncal) can not undermine the contribution of our framework. It only justify that both calibration and game-theoretic selction are indispensible part of our method. Specifically, our utility function estiamtion $\hat{U}$, entropy over all unlabeled nodes, is closer to ground-truth $U$, when prediction probability is accurate. That is why removing confidence calibration can lead to significant performance drop. Furthermore, Conf (Cal) is nearly equivelant to the best baseline methods CPL, but calibrated. It still cannot outperform our method in most datasets, and a case study is provided in Appendix E.2.
>
>
>
> > W3. The authors’ claim that "previous approaches all suffer from independent selection" (in introduction) may be overstated. For instance, CaGCN [1] propagates node confidence across the graph structure, reflecting combinatorial interactions through multi-hop neighbor consideration during calibration.
>
> **Response:** Thank you for pointing this out. We will improve the sentence as follows:
>
> *"Most of the previous approaches suffer from independent selection."*

---

> ### Author Response · Authors · 2024-11-20
>
> >Q1. CaGCN is not considered a SOTA calibration model. Could the authors compare the performance between BANGS and GATS [2]? While GATS does not include a self-training experiment, it could likely be adapted to a pseudo-labeling framework, similar to CaGCN.
> > Q2. Comparison with consistency regularization approaches, such as [3], is highly recommended, as the current baselines are all pseudo-labeling methods.
>
> **Response:** Thank you for useful suggestion. We have conducted the experiments as per your suggestions, and presented the results in Table 2, with best result in bold, and the second best italicised.
>
> We have adapted GATS [2] for self-training in our framework like Conf(Cal). The hyperparameters are set as the same as others. Though GATS is a more advanced technique than CaGCN and ETS, it performs well on Citeseer but does not outperform BANGS in other datasets. This again shows that, though calibrated confidence is an important part in our framework, our selection strategy plays a significant role as well.
> We also add the consistency regularization approach GRAND[3]. We use the default hyperparameters from their codebase in github. Our method outperforms GRAND in most of the settings.
>
> Table 2. Comparsion of our method with additional baselines.
>
> Dataset| CaGCN |  GATS | GRAND | BANGS|
> | -------- | -------- | -------- | ------- | -------|
> | Cora  | 83.06±0.11 | 83.76±0.18 | **85.13±0.21**| *84.02±0.13*|
> | Citeseer  |72.84±0.07  | **74.18±0.38** | *73.96±0.29*|  73.14±0.45|
> | PubMed  | *81.16±0.10*  |  80.25±0.37|  77.77±0.68|**81.60±0.34**|
> | LastFM  | 79.60±1.02  | 80.31±0.78 |  *82.21±1.05*| **83.27±0.48**|
> | Flickr  | 49.81±0.28 | *49.90±0.32* | 42.79±0.89| **50.23±0.25**|

---

> > ### Comment · Reviewer_oNSD · 2024-11-22
> > **Official Comment by Reviewer oNSD**
> >
> > Thanks for your detailed rebuttal. The authors have addressed most of my concerns, particularly regarding the necessity of employing a calibration model in BANGS. As such, I have decided to raise my score. I recommend the authors include this explanation in the ablation study in a future version of the manuscript.
> >
> > However, my concern about the efficiency still remains, especially on large-scale datasets. Nevertheless, I find this acceptable if the paper is appropriately cited, despite its somewhat limited practicality.

---

> > > ### Author Response · Authors · 2024-11-22
> > >
> > > Dear Reviewer,
> > >
> > > Thank you for acknowledging the contributions of our work! We will include the additional experiments in the ablation study as you have suggested.
> > >
> > > We understand your point. We will also add this discussion. As a future direction, we plan to focus on improving the efficiency of the utility function. One potential solution is through the use of influence functions, which, in current research, have been primarily explored on specific networks like GCNs [1,2]. We believe extending this approach to more complex GNNs could lead us to a more efficient framework.
> > >
> > > Thank you again for your valuable feedback.
> > >
> > > Best regards,
> > >
> > > The Authors
> > >
> > > [1] Chen, Z., Li, P., Liu, H., & Hong, P. Characterizing the Influence of Graph Elements. In The Eleventh International Conference on Learning Representations.
> > >
> > > [2] Kang, J., Zhou, Q., & Tong, H. (2022, August). JuryGCN: quantifying jackknife uncertainty on graph convolutional networks. In Proceedings of the 28th ACM SIGKDD Conference on Knowledge Discovery and Data Mining (pp. 742-752).

---

### Official Review · Reviewer_V2uF · 2024-11-03

**Soundness:** 2
**Presentation:** 3
**Contribution:** 2
**Rating:** 6
**Confidence:** 2

**Summary:**

To address the issue that existing pseudo-labeling strategy ignores the combinatorial dependencies between nodes, this paper introduces a framework that unifies the labeling strategy with conditional mutual information as the objective of node selection, instead of predicting pseudo labels.

**Strengths:**

1. It is interesting to introduce techniques from the co-operative game theory into graph self-tranining, taking into acount the combinatorial.
2. The paper is well-organized and literature review is conducted well.
3. The paper provides a theoretical analysis of the proposed method.

**Weaknesses:**

1. Considering the reported standard deviation of the experimental results, the improvements on some datasets appear to be marginal, such as on the Flickr dataset.
2. Experiments are conducted only on homophilic graph datasets. It would be better if the performance of the method on heterophilic graph datasets could be provided, such as Cornell, Texas, and Wisconsin [1].

[1] Geom-GCN: Geometric Graph Convolutional Networks, ICLR 2020

**Questions:**

Please refer to the Weaknesses.

---

> ### Author Response · Authors · 2024-11-20
>
> Thank you for the interesting comments and positive reviews.
>
> > 1. Considering the reported standard deviation of the experimental results, the improvements on some datasets appear to be marginal, such as on the Flickr dataset.
>
> **Response:** Thank you for pointing this out! Note that our method achieves best results in most of the setting.
>
> In the table below, we have performed additional experiments. We compare running time and test performance of our method with the raw model and best baseline, CPL, on the same setting as described in Section 4.1. The setting are the same as described in the main paper. We test on Flickr dataset with ~90k nodes and obgn-arxiv [a] dataset with ~170k nodes. We observe that graph self-training methods often exhibit comparatively smaller improvements on larger datasets. To address this problem, additional techniques, such as contrastive sampling [b], may be incorporated as a future direction.
>
> Running Time and Accuracy Comparsion of Flickr and obgn-arxiv Dataset.
>
> |  Dataset   |  Methods|Running Time(s)  | Accuracy (%)
> | --------| -------- | ------- | ----|
> | Flickr |  GCN | 15.16 | 49.53+-0.11|
> | Flickr |  CPL | 633.35 | 50.02+-0.22|
> | Flickr |  BANGS | 1189.48 | **50.23+-0.25** |
> | obgn-arxiv |  GCN | 30.81 | 63.82+-0.12 |
> | obgn-arxiv| CPL | 1287.57 | 63.81+-0.17 |
> | obgn-arxiv| BANGS | 1668.98 |  **63.86+-0.18** |
>
>
>
> [a] Wang, K., Shen, Z., Huang, C., Wu, C. H., Dong, Y., & Kanakia, A. (2020). Microsoft academic graph: When experts are not enough. Quantitative Science Studies, 1(1), 396-413.
>
> [b] Zhou, Z., Shi, J., Zhang, S., Huang, Z., & Li, Q. (2023). Effective stabilized self-training on few-labeled graph data. Information Sciences, 631, 369-384.
>
> > 2. Experiments are conducted only on homophilic graph datasets. It would be better if the performance of the method on heterophilic graph datasets could be provided, such as Cornell, Texas, and Wisconsin [1].
>
> **Response:** This is a good point. The current utility function is based on Personalized Pagerank (Definition 3.2), which operates on the assumption that importance is passed along edges. This assumption performs well in homophilic graphs, where similar nodes tend to cluster together. We will consider adapting our method to heterophilic graphs as an interesting future direction.

---

> > ### Comment · Reviewer_V2uF · 2024-12-03
> >
> > Thank you for your rebuttal. I appreciate the discussion you provided regarding my concern about the experimental results. After reviewing your explanations, I have decided to maintain my score.

---

### Official Review · Reviewer_g53d · 2024-11-04

**Soundness:** 3
**Presentation:** 4
**Contribution:** 3
**Rating:** 8
**Confidence:** 4

**Summary:**

The authors addressed the limitations in graph self-training by introducing a comprehensive framework that systematically tackles the node selection problem using a novel formulation with mutual information. The proposed BANGS is a game theory-based method with the
utility function based on feature propagation.

**Strengths:**

The writing is clear.

The discussion is comprehensive.

**Weaknesses:**

The novelty of the studied problem is limited - the authors studied the node selection problem using a novel formulation with mutual information.

The paper relies on a lot on (Wang & Jia, 2023) technically. It would be good to demonstrate the authors' unique contribution in the context of (Wang & Jia, 2023) - is it a straightforward extension of the referenced work?

**Questions:**

Given the improvement, how significant is the problem solved?

Is this work straightforward extension of the referenced work of (Wang & Jia, 2023)?

**Details Of Ethics Concerns:**

No concerns.

---

> ### Author Response · Authors · 2024-11-20
>
> Thank you for your positive reviews.
>
> > 1. The novelty of the studied problem is limited - the authors studied the node selection problem using a novel formulation with mutual information. Given the improvement, how significant is the problem solved?
>
> **Reponse:** Thank you for raising this point. Our proposed formulation is particularly well-suited for the graph node classification setting. Maximizing mutual information between predicted and ground-truth labels on unlabeled data is a natural and widely recognized objective in graph semi-supervised learning. Moreover, the loss function in graph node classification typically minimizes cross-entropy loss, which can be interpreted as a method to maximize our target: the mutual information between model predictions and true labels. Building on this formulation, we designed a utility function and introduced a scalable, effective approach to achieve this objective. Our experimental results demonstrate notable improvements, underscoring the significance of the problem addressed.
>
> > 2. The paper relies on a lot on (Wang & Jia, 2023) technically. It would be good to demonstrate the authors' unique contribution in the context of (Wang & Jia, 2023) - is it a straightforward extension of the referenced work? Is this work straightforward extension of the referenced work of (Wang & Jia, 2023)?
>
> **Response:** The main difference is that, we only consider coallitions of less than size $k$. This is tailered for our setting, as we select $k$ nodes in an iteration. Since we do not need to calculate utility of large coallitions, bounding size also significantly cut sampling cost.
>
> Further, the utility function is tailored for graph. The original utility function in (Wang & Jia, 2023) is the prediction accuracy on test dataset. This is extremely time-consuming, as it requires to add each coallition to psuedo-label set and retrain the model. Our utility function, as shown in Table 7 and 8 in our paper, allows the alogrithm to reach better performance within comparable time with other baselines.

---

### Official Review · Reviewer_ehCo · 2024-11-04

**Soundness:** 2
**Presentation:** 2
**Contribution:** 2
**Rating:** 5
**Confidence:** 3

**Summary:**

This paper focuses on the problem of graph self-training and argue that the combinatorial dependencies between nodes have a very strong relationship with the pseudo-labeling strategy which has been ignored now. To overcome this problem. this paper propose a novel framework that unifies the labeling strategy with conditional mutual information as the objective of node selection. This method grounded in game theory selects nodes in a combinatorial fashion and provides theoretical guarantees for robustness under noisy objective.

**Strengths:**

The structure of this article is clear and easy to understand. The experiments are detailed, and the analysis of the results is also quite clear. Moreover, the code has been made publicly available.

**Weaknesses:**

1. The motivation of the article is unclear and not strong enough. The core work of self-training is to select suitable nodes and assign pseudo-labels. This article focuses on the combinatorial dependencies in node selection; however, the impact of such information on the model's performance is not discussed in depth and lacks supporting experiments.

2. Furthermore, I do not see self-training as an interesting research direction; it seems more like a variant of data augmentation to me. Assigning labels to some nodes based on existing information intuitively seems difficult to understand, as it does not clearly lead to new information.

**Questions:**

1. The article has designed a k-Bounded Banzhaf value to measure the marginal contribution of  a node. Does it still satisfy the properties of the Banzhaf value, such as symmetry, additivity, and so on?
2. In Section 3.2 of the paper, the ground-truth utility function and the noisy utility function are mentioned. How is the ground-truth utility function calculated?

---

> ### Author Response · Authors · 2024-11-20
>
> Thank you for your detailed reviews and interesting points. We have clarified the points below.
>
> > 1. The motivation of the article is unclear and not strong enough. The core work of self-training is to select suitable nodes and assign pseudo-labels. This article focuses on the combinatorial dependencies in node selection; however, the impact of such information on the model's performance is not discussed in depth and lacks supporting experiments.
>
> **Response:** We agree that the core work of self-training is to select suitable nodes and assign pseudo-labels; and our goal is to select the optimal $k$ nodes. In order to select the suitable node set, dependencies between nodes needs to be considered. We have the followings in the paper to emphasize this point:
> * **Theoretical justification:** We provide a proof regarding why dependency must be considered to select optimal nodes in Appendix B.2.
> * **Emperical Justification:** Moreover, we provide ablation studies in Table 2. Without using Banzhaf value, we observe a significant performance drop for the proposed algorithm. This can jusify the importance of considering combinatorial dependencies in node selection.
>
> Before going to details of our method (Sec 3.1), to emphasize the motivation we will add a discussion based on the above as well as the answer for the next point.
>
> > 2.Furthermore, I do not see self-training as an interesting research direction; it seems more like a variant of data augmentation to me. Assigning labels to some nodes based on existing information intuitively seems difficult to understand, as it does not clearly lead to new information.
>
> **Response:** This is a good point. There is a fundamental difference between self-training and data augmentation. Though both can improve model robustness and generalization, self-training is not a variant of data augmentation. Data augmentation focuses on generating new variations of the existing labeled data. Self-training focuses on leveraging the unlabeled data to make the model more effective by adding new labeled examples through its own predictions. There is also research which proves that self-taining can improve upon the initial model with sufficicient unlabeled data [1]. Empirically, we have observed that the prediction accuracy on the test set exhibits an increasing trend across self-training iterations (see Figure 4 in the Appendix).
>
> [1] Zhang, S., Wang, M., Liu, S., Chen, P. Y., & Xiong, J. (2022, April). How does unlabeled data improve generalization in self-training? A one-hidden-layer theoretical analysis. In International Conference on Learning Representations.
>
>
> > 3. The article has designed a k-Bounded Banzhaf value to measure the marginal contribution of a node. Does it still satisfy the properties of the Banzhaf value, such as symmetry, additivity, and so on?
>
> **Response:** Banzhaf value is a semivalue. The only difference between Banzhaf value and our adapted $k$-bounded version one is that we only allow colliations less than size $k$. We still satisfy dummy player, symmetry and linerity as the original Banzhaf value.
>
> > 4. In Section 3.2 of the paper, the ground-truth utility function and the noisy utility function are mentioned. How is the ground-truth utility function calculated?
>
> **Response:** Thank you for pointing this out. The ground-truth utility is the information gain (reduced uncertainty) over all unlabeled nodes with ground-truth student model logits $\tilde{X}$, i.e., $\hat{U}$(S) = O(Softmax($\tilde{X}$(S))), similar to Equation 4.

---

> > ### Comment · Reviewer_ehCo · 2024-11-22
> >
> > Thank you for the author's reply. In fact, the author did not directly address my concerns, but instead provided evidence from another perspective to support the significance of the research in this paper, which I am not very satisfied with. I am very puzzled to see that some reviewers gave 8 points. This article is at most a borderline article, and giving 8 points is too high. Based on the above, I will keep my score.

---

> > > ### Author Response · Authors · 2024-11-22
> > >
> > > Dear Reviewer,
> > >
> > > Thank you for your feedback. We are unclear about the specific concerns that you feel have not been addressed. Could you kindly provide more details or clarify your expectations so that we can better address your concerns?
> > >
> > > We greatly appreciate any additional comments or suggestions you may have, as they will help us further improve the paper. We believe self-training is an important area of research in semi-supervised learning. In our work, we contribute to the field of graph self-training by introducing a novel, effective, and theoretically grounded method that leverages combinatorial dependencies for improved node selection in graph self-training.
> > >
> > > Thank you again for your valuable time and feedback.
> > >
> > > Best Regards,
> > >
> > > Authors

---

> > > > ### Author Response · Authors · 2024-12-03
> > > >
> > > > Dear Reviewer ehCo,
> > > >
> > > > As the discussion phase is nearing its conclusion, we would sincerely appreciate it if you could let us know your specific concerns. Thank you once again for your time.
> > > >
> > > > Best regards,
> > > >
> > > > Authors

---

### Official Review · Reviewer_A5dG · 2024-11-04

**Soundness:** 3
**Presentation:** 3
**Contribution:** 3
**Rating:** 6
**Confidence:** 4

**Summary:**

This paper investigates the problem of graph self-training, which is a main strategy in semi-supervised graph learning. The study specifically addresses the combinatorial dependencies between nodes for pseudo-label selection. To tackle this challenge, the authors propose a novel framework that unifies the labeling strategy with conditional mutual information to guide the selection of pseudo-labels. Unlike traditional approaches that provide a sorted list of labels, the proposed method forms a node set for graph self-training. The proposed method is validated on many real-world datasets.

---
After rebuttal, I raise my rating from 5 to "6: marginally above the acceptance threshold."

**Strengths:**

This paper introduces a new direction in graph self-training by integrating conditional mutual information into the pseudo-labeling process. The proposed method may have the potential to improve the effectiveness of semi-supervised learning on graphs, particularly in scenarios where node dependencies play a significant role. The empirical studies are solid.

**Weaknesses:**

1. One main concern is the rationale behind forming a node set for graph self-training from a submodular optimization perspective. The paper argues that pseudo-labels should be evaluated and fed into the model as a set, contrasting with most existing self-training strategies that evaluate each pseudo-label individually. The justification of the traditional strategy is that adding pseudo-labels to the training set satisfies submodularity, allowing for the use of a greedy strategy to achieve an optimal solution.

Does submodular optimization apply to the formation of the node set in this paper? Why this approach is advantageous?

2. In the experiments, the number of pseudo-labels $k$ is fixed. This setting may not be fair for comparisons across different methods. Allowing each baseline to obtain an optimal number of pseudo-labels as they required would provide a more equitable evaluation and potentially yield more insightful results.

**Questions:**

See the weaknesses.

---

> ### Author Response · Authors · 2024-11-20
>
> Thank you for your excellent point about submodularity.
>
> > 1.One main concern is the rationale behind forming a node set for graph self-training from a submodular optimization perspective. The paper argues that pseudo-labels should be evaluated and fed into the model as a set, contrasting with most existing self-training strategies that evaluate each pseudo-label individually. The justification of the traditional strategy is that adding pseudo-labels to the training set satisfies submodularity, allowing for the use of a greedy strategy to achieve an optimal solution.Does submodular optimization apply to the formation of the node set in this paper? Why this approach is advantageous?
>
> **Response:** Thank you for this interseting question. Unfortunately, our objective function does not satisfy submodularity property. In graph learning, the logits are predicted by GNN fitted on node features, labels, and graph structure.
>
> Submodularity requires that for each $S \subset T$ and every node $v \notin T$, $U(S \cup v)-U(S) > U(T \cup v)-U(T)$. We can provide a simple counter example to prove that the objective function is not submodular. Consider a graph with node $v_1$,$v_2$,$v_3$,$v_4$, and edge between $(v_1,v_2)$, $(v_2,v_3)$, and $(v_3,v_4)$. Node $v_2$ have two neighbors, Node $v_1$ and $v_3$. For simplicity, consider binary classification, node $v_1$, $v_2$, $v_4$ belong to class 0, and $v_3$ is class 1. Suppose an initial model can predict each node to the right class with 0.8. Let $S = \varnothing$, $T = \{v_1\}$, and $v = v_3$. When adding $v_3$ of class 1, the uncertainty over all other nodes will increase. This is because most GNNs have homophily assumption, and the other nodes are from a different class as $v_3$. Therefore, $U(S \cup v)-U(T)$ will be negative. When we have $T = \{v_1\}$, adding a $v_3$ from class 1 would still raise uncertainty over the unlabeled $v_2$ and $v_4$. But since GNNs aggregate information from neighbors, the influence from $v_3$ of different class is partially offset by $v_1$ of the correct class. Therefore, $U(T \cup v)-U(T) > U(S \cup v)-U(S)$.
>
> Since the objective function is not submodular, our combinatorial solution is more beneficial than independent selection, and the proof is provided in Appendix B.2.
>
>
> > 2. In the experiments, the number of pseudo-labels k is fixed. This setting may not be fair for comparisons across different methods. Allowing each baseline to obtain an optimal number of pseudo-labels as they required would provide a more equitable evaluation and potentially yield more insightful results.
>
>
> **Response:** To clarify, we have chosen the optimal number of $k$ for baseline methods according to the suggestions provided in the corresponding papers. Current methods either fix the number of nodes selected, or determine the confidence threshold, i.e., selecting nodes with confidence surpassing a given threshold into pseudo-labeled set in each iteration.
> * Baseline M3S and DR-GST do not explicitly provide their hyperparameters, so we use their configurations implemented by [1]. The threshold is predetermined for each dataset, e.g., 0.5 for Cora, 0.2 for Citeseer.
> * Similarly, for CaGCN, We use their suggested threshold, e.g., 0.8 for Cora, 0.9 for Citeseer. CPL specifies their number of selected nodes, e.g., top 100 for Cora, top 80 for Citeseer.
>
> In conclusion, we belive we have a fair comparion with the baselines with their suggested optimal parameters.
>
> [1] WANG Botao, Jia Li, Yang Liu, Jiashun Cheng, Yu Rong, Wenjia Wang, and Fugee Tsung. Deep insights into noisy pseudo labeling on graph data. In Thirty-seventh Conference on Neural Information Processing Systems, 2023.

---

> ### Author Response · Authors · 2024-11-25
>
> Dear Reviewer A5dG,
>
> We would like to kindly follow up and reiterate that we have responded to your concerns regarding submodularity and the problem setting. Specifically, we clarified that our problem is non-submodular. Additionally, we have ensured that all baselines adopt the optimal setting with respect to the number of $k$.
>
> As the discussion phase is nearing its conclusion, we would sincerely appreciate it if you could let us know whether our clarifications and rebuttal have satisfactorily addressed your concerns.
>
> Thank you once again for your valuable time and feedback.
>
> Best regards,
>
> Authors

---

> ### Comment · Reviewer_A5dG · 2024-12-02
> **Thanks for your reply**
>
> I still believe that adding pseudo-labels satisfies submodularity in contributing useful information to the model. In other words, incrementally adding pseudo-labels using a greedy strategy is optimal. The issues highlighted in this paper, as you mentioned, likely stem from the imbalanced quantity or distribution of labels, which introduces additional challenges in model training. These challenges could potentially be addressed through constraints or simple adjustments.
>
> In summary, if the relevant statements can be clearly clarified, particularly the reason behind “pseudo-labels should be fed into the model as a set”, I think this paper holds merit and would like to increase my rating score.

---

> > ### Author Response · Authors · 2024-12-02
> >
> > We sincerely appreciate your comment! Thank you!
> >
> >
> > First of all, we clarify that the current iterative graph psuedo-labeling commonly adopts greedy strategy. In each iteraction, all baselines and our method are selecting the best nodes and retrain the model. However, within each iteration, how to select nodes can vary, and most methods follow greedy strategy according to some functions, e.g., the confidence of single node.
> >
> > > I still believe that adding pseudo-labels satisfies submodularity in contributing useful information to the model. In other words, incrementally adding pseudo-labels using a greedy strategy is optimal.
> >
> > Secondly, we believe the submodularity property in this problem is related to the selection of function U, as defined in the case study of our last response. It is *NOT* submodular for arbitrary common functions, e.g., confidence of a single node, or information gain from a node over all unlabeled nodes. The key is that selection and information between nodes are interdepent, and previous measures contain either insufficient or overlapping information. For the former measure, we have proven this via a counterexample in the previous response. In Table 2 of ablation study, we also provide the results of BANGS (No Banzhaf) that selects the informative nodes greedily, and the performance significantly drops.
> >
> > It may be possible to convert this problem into submodular with an appropriate function, which can dynamically update the value after selecting each psuedo-label and behave as a coverage function. However, even if it is submodular, this function is hard to design and estimate, and a corresponding greedy algorithm will be less robust to noise. We also humbly point out that submodularity does not imply that the greedy algorithm will be optimal, but guarantees that greedy has a constant factor approximation [1].
> >
> > > The issues highlighted in this paper, as you mentioned, likely stem from the imbalanced quantity or distribution of labels, which introduces additional challenges in model training. These challenges could potentially be addressed through constraints or simple adjustments.
> >
> > This is an excellent point. We have included a baseline, DR-GST, that recognizes the distribution shift problem and adjusts the self-training optimization goal to counter the effect. In Table 1 and Table 3, this method cannot outperform our method across different datasets and base models. This means that these challanges might *NOT* be easy to address through simple adjustments. Banzhaf value is a more effective way to provide fair allocation for the contribution to nodes.
> >
> > We will add this discussion in the next version of our paper.
> >
> >
> > [1] G. L. Nemhauser and L. A. Wolsey. Best algorithms for approximating the maximum of a submodular set function. Mathematics of operations research, 3(3):177–188, 1978.

---

### Author Response · Authors · 2024-11-20

We sincerely thank the reviewers for their valuable feedback, insightful comments, which have helped us further clarify and enhance our work. A comprehensive point-by-point response to the reviewers' comments is presented below.  The *major additional changes* are listed below.

* **Additional experiments:** We have incorporated all of the additional experiments requested by the reviewers. These include running time and performance on larger graphs, and comparisons with additional baseline methods.
* **Other discussions:** We have undertaken a series of discussions, which include the following aspects:
    1. The non-submodularity nature and the necessity of considering dependencies in the node selection problem.
    2. Clarifications on experimental settings, the role of the calibration method, and other details.
    3. Discussion on extended application in other settings, e.g., large-scale graphs and heterophilic graphs.

We hope these will satisfactorily address the concerns raised by the reviewers and elevate the overall quality of our work.

---

### Meta-Review · Area_Chair_j7gu · 2024-12-21

**Metareview:**

The paper proposes BANGS, a novel framework that unifies the labeling strategy with conditional mutual information for graph self-training. The problem is interesting. The experiments are comprehensive and solid. Sufficient theoretical analysis is provided. The presentation is good. Several issues need to be addressed to improve the paper, such as unclear motivation, a lack of sufficient discussion of existing work (Wang & Jia, 2023), and marginal improvement. Reviewers are generally positive about this work.

**Additional Comments On Reviewer Discussion:**

There is one reviewer who gave a relatively negative score. All other reviewers are positive about this work.

---

### Decision · Program_Chairs · 2025-01-22

Accept (Poster)